# Learning to Generate Realistic Noisy Images via Pixel-level Noise-aware Adversarial Training

**Yuanhao Cai** [1,2], **Xiaowan Hu** [1,2], **Haoqian Wang** [1,2,*],
**Yulun Zhang** [3], **Hanspeter Pfister** [4], **Donglai Wei** [5]
[1] Shenzhen International Graduate School, Tsinghua University,
[2] Shenzhen Institute of Future Media Technology,
[3] ETH Zürich, [4] Harvard University, [5] Boston College

## Abstract

Existing deep learning real denoising methods require a large amount of noisy-clean image pairs for supervision. Nonetheless, capturing a real noisy-clean dataset is an unacceptable expensive and cumbersome procedure. To alleviate this problem, this work investigates how to generate realistic noisy images. Firstly, we formulate a simple yet reasonable noise model that treats each real noisy pixel as a random variable. This model splits the noisy image generation problem into two sub-problems: image domain alignment and noise domain alignment. Subsequently, we propose a novel framework, namely Pixel-level Noise-aware Generative Adversarial Network (PNGAN). PNGAN employs a pre-trained real denoiser to map the fake and real noisy images into a nearly noise-free solution space to perform image domain alignment. Simultaneously, PNGAN establishes a pixel-level adversarial training to conduct noise domain alignment. Additionally, for better noise fitting, we present an efficient architecture Simple Multi-scale Network (SMNet) as the generator. Qualitative validation shows that noise generated by PNGAN is highly similar to real noise in terms of intensity and distribution. Quantitative experiments demonstrate that a series of denoisers trained with the generated noisy images achieve state-of-the-art (SOTA) results on four real denoising benchmarks.

## 1 Introduction

Image denoising is an important yet challenging problem in low-level vision. It aims to restore a clean image from its noisy counterpart. Traditional approaches concentrate on designing a rational maximum *a posteriori* (MAP) model, containing regularization and fidelity terms, from a Bayesian perspective [1]. Some image priors like low-rankness [2, 3, 4], sparsity [5], and non-local similarity [6, 7] are exploited to customize a better rational MAP model. However, these hand-crafted methods are inferior in representing capacity. With the development of deep learning, image denoising has witnessed significant progress. Deep convolutional neural network (CNN) applies a powerful learning model to eliminate noise and has achieved promising performance [8, 9, 10, 11, 12, 13, 14]. These deep CNN denoisers rely on a large-scale dataset of real-world noisy-clean image pairs. Nonetheless, collecting even small datasets is extremely tedious and labor-intensive. The process of acquiring real-world noisy-clean image pairs is to take hundreds of noisy images of the same scene and average them to get the clean image. To get more image pairs, researchers try to synthesize noisy images.

In particular, there are two common settings for synthesizing noisy images. As shown in Fig. 1 (a1), setting1 directly adds the additive white Gaussian noise (AWGN) with the clean RGB image. For a long time, single image denoising [15, 16, 17, 18, 19, 10] is performed with setting1. Nevertheless, fundamentally different from AWGN, real camera noise is generally more sophisticated and signal-dependent[20, 21]. The noise produced by photon sensing is further affected by the in-camera signal processing (ISP) pipeline (e.g., Gama correction, compression, and demosaicing). Models

---

*[*]Haoqian Wang is the corresponding author, email: wanghaoqian@tsinghua.edu.cn*

35th Conference on Neural Information Processing Systems (NeurIPS 2021).

trained with setting1 are easily over-fitted to AWGN and fail in real noise removal. Setting2 is based on ISP-modeling CNN [22] and Poisson-Gaussian [21, 23] noise model that modeling photon sensing with Poisson and remaining stationary disturbances with Gaussian has been adopted in RAW denoising. As shown in Fig. 1 (a2), setting2 adds a Poisson-Gaussian noise with the clean RAW image and then passes the result through a pre-trained RAW2RGB CNN to obtain the RGB noisy counterpart. Notably, when the clean RAW image is unavailable, a pre-trained RGB2RAW CNN is utilized to transform the clean RGB image to its RAW counterpart [22]. However, setting2 has the following drawbacks: **(i)** The noise is assumed to obey a hand-crafted probability distribution. However, because of the randomness and complexity of real camera noise, it's difficult to customize a hand-crafted probability distribution to model all the characteristics of real noise. **(ii)** The ISP pipeline is very sophisticated and hard to be completely modeled. The RAW2RGB branch only learns the mapping from the clean RAW domain to the clean RGB space. However, the mapping from the Poisson-Gaussian noisy RAW domain to the real noisy RGB space can not be ensured. **(iii)** The ISP pipelines of different devices vary significantly, which results in the poor generality and robustness of ISP modeling CNNs. Thus, whether noisy images are synthesized with setting1 or 2, there still remains a discrepancy between synthetic and real noisy datasets. We notice that GAN utilizes the internal information of the input image and external information from other images when modeling image priors. Hence, we propose to use GAN to adaptively learn the real noise distribution.

GAN is firstly introduced in [24] and has been proven successful in image synthesis [25, 26, 27] and translation [26, 27]. Subsequently, GAN is applied to image restoration and enhancement, e.g., super resolution [28, 29, 30], style transfer [27, 31], enlighten [32, 33], deraining [34], dehazing [35], image inpainting [36, 37], image editing [38, 39], and mobile photo enhancement [40, 41]. Although GAN is widely applied in low-level vision tasks, few works are dedicated to investigating the realistic noise generation problem [42]. Chen et al. [43] propose a simple GAN that takes Gaussian noise as input to generate noisy patches. However, as in general, this GAN is image-level, i.e., it treats images as samples and attempts to approximate the probability distribution of real-world noisy images. This image-level GAN neglects that each pixel of a real noisy image is a random variable and the real noise is spatio-chromatically correlated, thus results in coarse learning of the real noise distribution.

To alleviate the above problems, this work focuses on learning how to generate realistic noisy images so as to augment the training data for real denoisers. To begin with, we propose a simple yet reasonable noise model that treats each pixel of a real noisy image as a random variable. This noise model splits the noise generation problem into two sub-problems: image domain alignment and noise domain alignment. Subsequently, to tackle these two sub-problems, we propose a novel Pixel-level Noise-aware Generative Adversarial Network (PNGAN). During the training procedure of PNGAN, we employ a pre-trained real denoiser to map the generated and real noisy images into a nearly noise-free solution space to perform image domain alignment. Simultaneously, PNGAN establishes a pixel-level adversarial training that encourages the generator to adaptively simulate the real noise distribution so as to conduct the noise domain alignment. In addition, for better real noise fitting, we present a lightweight yet efficient CNN architecture, Simple Multi-scale Network (SMNet) as the generator. SMNet repeatedly aggregates multi-scale features to capture rich auto-correlation, which provides more sufficient spatial representations for noise simulating. Different from general image-level GAN, our discriminator is pixel-level. The discriminator outputs a score map. Each position on the score map indicates how realistic the corresponding noisy pixel is. With this pixel-level noise-aware adversarial training, the generator is encouraged to create solutions that are highly similar to real noisy images and thus difficult to be distinguished.

In conclusion, our contributions can be summarized into four points:

(1) We formulate a simple yet reasonable noise model. This model treats each noisy pixel as a random variable and then splits the noisy image generation into two parts: image and noise domain alignment.

(2) We propose a novel framework, PNGAN. It establishes an effective pixel-level adversarial training to encourage the generator to favor solutions that reside on the manifold of real noisy images.

(3) We customize an efficient CNN architecture, SMNet learning rich multi-scale auto-correlation for better noise fitting. SMNet serves as the generator in PNGAN costing only 0.8M parameters.

(4) Qualitative validation shows that noise generated by PNGAN is highly similar to real noise in terms of intensity and distribution. Quantitative experiments demonstrate that a series of denoisers finetuned with the generated noisy images achieve SOTA results on four real denoising benchmarks.

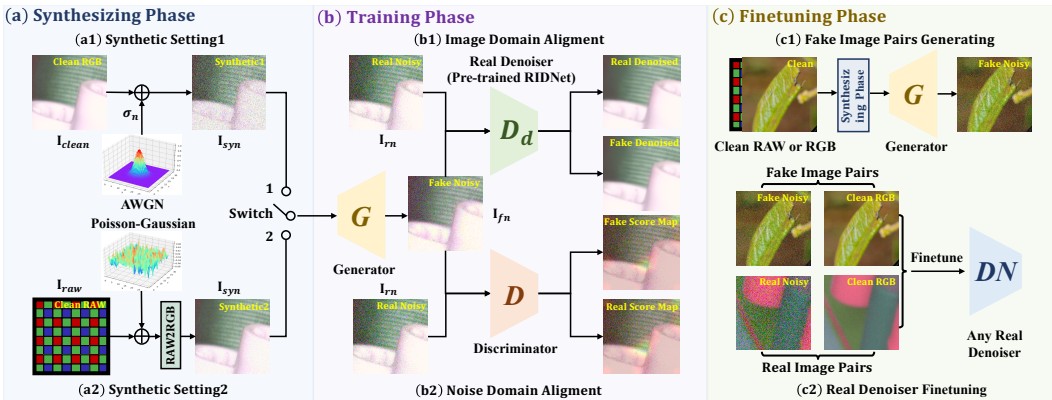

Figure 1: The pipeline of using PNGAN to perform data augmentation. It is divided into: (a) synthesizing phase, (b) training phase, and (c) finetuning phase. Please refer to the text (Sec. 2) for more detailed descriptions.

## 2  Proposed Method

As shown in Fig. 1, the pipeline of using PNGAN to perform data augmentation consists of three phases. (a) is the synthesizing phase. (a1) and (a2) are two common synthetic settings. In this phase, we produce the synthetic noisy image from its clean RGB or RAW counterpart. (b) is the training phase of PNGAN. The generator $G$ adopts the synthetic image as input. Which synthetic setting is selected is controlled by the switch. By using a pre-trained real denoiser $D_d$, PNGAN establishes a pixel-level noise-aware adversarial training between the generator $G$ and discriminator $D$ so as to simultaneously conduct image and noise domain alignment. $D_d$ is set as RIDNet [44] in this work. (c) is the finetuning phase. Firstly, in (c1), the generator creates extended fake noisy-clean image pairs. Secondly, in (c2), the fake and real data are jointly utilized to finetune a series of real denoisers.

### 2.1  Pixel-level Noise Modelling

Real camera noise is sophisticated and signal-dependent. Specifically, in the real camera system, the RAW noise produced by photon sensing comes from multiple sources (e.g., short noise, thermal noise, and dark current noise) and is further affected by the ISP pipeline. Besides, illumination changes and camera movement inevitably lead to spatial pixel misalignment and color or brightness deviation. Hence, hand-designed noise models based on mathematical assumptions are difficult to accurately and completely describe the properties of real noise. Different from previous methods, we don't base our noise model on any mathematical assumptions. Instead, we use CNN to implicitly simulate the characteristics of real noise. We begin by noting that when taking multiple noisy images of the same scene, the noise intensity of the same pixel varies a lot. Simultaneously, affected by the ISP pipeline, the real noise is spatio-chromatically correlated. Thus, the correlation between different pixels of the same real noisy image should be considered. In light of these facts, we treat each pixel of a real noisy image as a random variable and formulate a simple yet reasonable noise model:

$$\mathbf{I}_{rn}[i] = \hat{\mathbf{I}}_{clean}[i] + \mathbf{N}[i], \quad D_d(\mathbf{I}_{rn})[i] = \hat{\mathbf{I}}_{clean}[i], \quad 1 \le i \le H \times W, \tag{1}$$

where $\hat{\mathbf{I}}_{clean} \in \mathbb{R}^{H \times W \times 3}$ is the predicted clean counterpart of $\mathbf{I}_{rn}$, it's denoised by $D_d$. Each $\mathbf{N}[i]$ is a random noise variable with unknown probability distribution. Therefore, each $\mathbf{I}_{rn}[i]$ can also be viewed as a distribution-unknown random variable. Now we aim to design a framework to generate a fake noisy image $\mathbf{I}_{fn} \in \mathbb{R}^{H \times W \times 3}$ such that the probability distribution of $\mathbf{I}_{fn}[i]$ and $\mathbf{I}_{rn}[i]$ is as close as possible. Please note that the mapping learned by $D_d$ is precisely from $\mathbf{I}_{rn}$ to $\hat{\mathbf{I}}_{clean}$. If the constant in Eq. (1) is set as the clean image $\mathbf{I}_{clean} \in \mathbb{R}^{H \times W \times 3}$, the subsequent domain alignment will introduce unnecessary errors and eventually lead to inaccurate results.

### 2.2  Pixel-level Noise-aware Adversarial Training

Our goal is to generate realistic noisy images. According to the noise model in Eq. (1), we split this problem into two sub-problems: (i) Image domain alignment aims to align $\hat{\mathbf{I}}_{clean}[i]$. (ii) Noise domain alignment targets at modeling the distribution of $\mathbf{N}[i]$. To handle the sub-problems, PNGAN establishes a novel pixel-level noise-aware adversarial training between $G$ and $D$ in Fig. 1 (b).

**Image Domain Alignment.** A very naive strategy to construct both image and noise domain alignment is to directly minimize the distance of $\mathbf{I}_{fn}$ and $\mathbf{I}_{rn}$. However, due to the intrinsic randomness, complexity, and irregularity of real noise, directly deploying $\mathcal{L}_1$ loss between $\mathbf{I}_{fn}$ and $\mathbf{I}_{rn}$ is unreasonable and drastically damages the quality of $\mathbf{I}_{fn}$. Besides, as analyzed in Sec. 2.1, each pixel of $\mathbf{I}_{rn}$ is a distribution-unknown random variable. This indicates that such a naive strategy challenges the training and may easily cause the non-convergence issue. Therefore, the noise interference should be eliminated while constructing the image domain alignment. To this end, we feed $\mathbf{I}_{fn}$ and $\mathbf{I}_{rn}$ into $D_d$ to obtain their denoised versions and then perform $\mathcal{L}_1$ loss between $\mathbf{I}_{fn}$ and $\mathbf{I}_{rn}$:

$$\mathcal{L}_1 = \sum_{i=1}^{H \times W} \big|\big| D_d(\mathbf{I}_{fn})[i] - D_d(\mathbf{I}_{rn})[i] \big|\big|_1 = \sum_{i=1}^{H \times W} \big|\big| D_d(\mathbf{I}_{fn})[i] - \hat{\mathbf{I}}_{clean}[i] \big|\big|_1. \quad (2)$$

By using $D_d$, we can transfer $\mathbf{I}_{rn}$ and $\mathbf{I}_{fn}$ into a nearly noise-free solution space. The value of $\hat{\mathbf{I}}_{clean}$ is relatively stable. Therefore, minimizing $\mathcal{L}_1$ can encourage $G$ to favor solutions that after being denoised by $D_d$ converge to $\hat{\mathbf{I}}_{clean}$. In this way, the image domain alignment is constructed.

**Noise Domain Alignment.** Becasue of the complexity and variability of real noise, it's hard to completely seperate $\mathbf{N}[i]$ from $\mathbf{I}_{rn}[i]$ in Eq (1). Fortunately, we note that on the basis of constructing the image domain alignment of $\hat{\mathbf{I}}_{clean}[i]$, the noise domain alignment of $\mathbf{N}[i]$ is equivalent to the distribution estimation of $\mathbf{I}_{rn}[i]$. Additionally, as the real noise is signal-dependent, the alignment between $\mathbf{I}_{fn}[i]$ and $\mathbf{I}_{rn}[i]$ is more beneficial to capture the correlation between noise and scene. We denote the

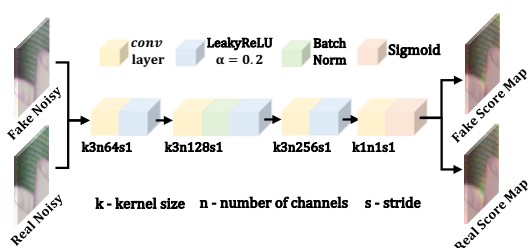

Figure 2: Architecture of discriminator.

distribution of $\mathbf{I}_{rn}[i]$ as $P_{data}(x_i)$, some real noisy pixel samples of $\mathbf{I}_{rn}[i]$ as $\{x_i^1, x_i^2, ..., x_i^m\}$ such that $x_i^k \sim P_{data}(x_i)$, and the distribution of $\mathbf{I}_{fn}[i]$ as $P_G(x_i; \theta_G)$. Here $\theta_G$ is the parameter of $G$. Then we formulate the noise domain aligment into a maximum likelihood estimation problem:

$$\theta_G^* = \underset{\theta_G}{arg\,max} \sum_{i=1}^{H \times W} \sum_{k=1}^{m} \log P_G(x_i^k; \theta_G) = \underset{\theta_G}{arg\,max}\, \mathbb{E}_i \big[\, \mathbb{E}_{x_i^k} \big[\, \log P_G(x_i^k; \theta_G) \,\big] \,\big], \quad (3)$$

where $\mathbb{E}$ means taking the average value. To approach this upper bound as close as possible, we present $D$ and establish the pixel-level adversarial traininig between $G$ and $D$. The architecture of $D$ is shown in Fig. 2. $D$ consists of 4 convolutional ($conv$) layers and utilizes LeakyReLU activation ($\alpha = 0.2$). General discriminator treats a image as a sample and outputs a score indicating how realistic the image is. Instead, $D$ is a pixel-level classifier. $D$ adopts the fake and real noisy images as input in a mini-batch and outputs a score map $\mathbf{P} \in \mathbb{R}^{H \times W}$ for each image. Specifically, the information of $\mathbf{P}[i] \in [0, 1]$ is the probability value indicating how realistic $P_G(x_i; \theta_G)$ is. $G$ aims to generate more realistic $\mathbf{I}_{fn}[i]$ to fool $D$ while $D$ targets at distinguishing $\mathbf{I}_{fn}[i]$ from $\mathbf{I}_{rn}[i]$. According to Eq .(3), we formulate the adversarial training between $G$ and $D$ as a min-max problem:

$$\underset{\theta_G}{min}\, \underset{\theta_D}{max}\; \mathbb{E}_i \big[\mathbb{E}_{\mathbf{I}_{rn}} \big[\log(D(\mathbf{I}_{rn}; \theta_D)[i])\big]\big] + \mathbb{E}_i \big[\mathbb{E}_{\mathbf{I}_{fn}} \big[\log(1 - D(\mathbf{I}_{fn}; \theta_D)[i])\big]\big], \quad (4)$$

where $\mathbb{E}_{\mathbf{I}_{rn}}$ and $\mathbb{E}_{\mathbf{I}_{fn}}$ respectively represent the operation of taking the average for all fake and real data in the mini-batch. As analyzed in [45], to make GANs analogous to divergence minimization and produce sensible predictions based on the *a priori* knowledge that half of the samples in the mini-batch are fake, we utilize the recently proposed relativistic discriminator [45] as follow:

$$D(\mathbf{I}_{rn}; \theta_D) = \sigma(C_D(\mathbf{I}_{rn})), \quad D_{Ra}(\mathbf{I}_{rn}, \mathbf{I}_{fn}) = \sigma(C_D(\mathbf{I}_{rn}) - \mathbb{E}_{\mathbf{I}_{fn}}(C_D(\mathbf{I}_{fn}))),$$
$$D(\mathbf{I}_{fn}; \theta_D) = \sigma(C_D(\mathbf{I}_{fn})), \quad D_{Ra}(\mathbf{I}_{fn}, \mathbf{I}_{rn}) = \sigma(C_D(\mathbf{I}_{fn}) - \mathbb{E}_{\mathbf{I}_{rn}}(C_D(\mathbf{I}_{rn}))), \quad (5)$$

where $D_{Ra}$ denotes the relativistic discriminator, $\sigma$ means the Sigmoid activation, and $C_D$ represents the non-transformed discriminator output. $D_{Ra}$ estimates the probability that real data is more realistic than fake data and also directs the generator to create a fake image that is more realistic than real images. The loss functions of $D$ and $G$ are then defined in a symmetrical form:

$$\mathcal{L}_D = -\mathbb{E}_i \big[\mathbb{E}_{\mathbf{I}_{rn}}[\log(D_{Ra}(\mathbf{I}_{rn}, \mathbf{I}_{fn})[i])] + \mathbb{E}_{\mathbf{I}_{fn}}[\log(1 - D_{Ra}(\mathbf{I}_{fn}, \mathbf{I}_{rn})[i])]\big],$$
$$\mathcal{L}_G = -\mathbb{E}_i \big[\mathbb{E}_{\mathbf{I}_{rn}}[\log(1 - D_{Ra}(\mathbf{I}_{rn}, \mathbf{I}_{fn})[i])] + \mathbb{E}_{\mathbf{I}_{fn}}[\log(D_{Ra}(\mathbf{I}_{fn}, \mathbf{I}_{rn})[i])]\big]. \quad (6)$$

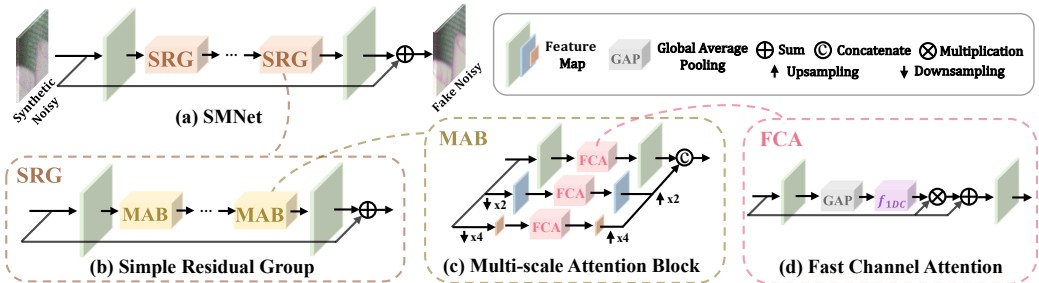

Figure 3: Details of the generator. (a) is the architecture of SMNet. (b) depicts the components of SRG. (c) shows the details of MAB. MAB is equipped with FCA, which is illustrated in (d).

During the training procedure, we fix $D$ to train $G$ and fix $G$ to train $D$ iteratively. Minimizing $\mathcal{L}_G$ and $\mathcal{L}_D$ alternately allows us to train a generative model $G$ with the goal of fooling the pixel-level discriminator $D$ that is trained to distinguish fake noisy images from real noisy images. This pixel-level noise-aware adversarial training scheme encourages $G$ to favor perceptually natural solutions that reside on the manifold of real noisy images so as to construct the noise domain alignment.

## 2.3 Noisy Image Generating

In Sec. 2.1, we denote the probability distribution of $\mathbf{I}_{fn}[i]$ as $P_G(x_i; \theta_G)$. Now we customize a light-weight yet efficient CNN architecture, SMNet as $G$ to generate $P_G(x_i; \theta_G)$. In this section, we firstly introduce the input setting of $G$ and subsequently detail the architecture of SMNet.

**Input Setting.** We aim to generate a realistic noisy image from its clean counterpart. A naive setting is to directly adopt the clean image as the input to generate the noisy image. However, this naive setting is not in line with the fact. When we repeatedly feed the same clean image to a pre-trained $G$, $G$ outputs completely the same noisy images. In contrast, when taking multiple pictures in the real world, the real noisy images vary a lot in the intensity of each pixel. This is caused by many factors (e.g., photon sensing noise, ISP pipelines, and illumination conditions). Hence, the naive input setting containing no distribution is unreasonable. We review that the general GANs sample from an initial random distribution (usually Gaussian) to generate a fake image. Hence, the input of $G$ should contain a random distribution so as to generate multiple noisy images of the same scene. We note that the two common synthetic settings meet this condition. Therefore, we utilize the two common settings to produce the synthetic image and then adopt the synthetic image as the input of $G$. Subsequently, we propose a light-weight yet efficient architecture, SMNet for better real noise fitting.

**SMNet Architecture.** The architecture of SMNet is shown in Fig. 3 (a). SMNet involves $t$ Simple Residual Groups (SRG) and each SRG contains $n$ Multi-scale Attention Blocks (MAB). The synthetic input $\mathbf{I}_{syn} \in \mathbb{R}^{H \times W \times 3}$ continuously undergoes a *conv* layer $f_1$, $t$ SRGs, and a *conv* layer $f_2$, then adds with a long identity mapping for efficient residual learning to eventually generate the fake noisy counterpart $\mathbf{I}_{fn} \in \mathbb{R}^{H \times W \times 3}$. This process can be formulated as:

$$\mathbf{I}_{fn} = \mathbf{I}_{syn} + f_2(S_t(\mathbf{F}_{S_t})), \quad \mathbf{F}_{S_{j+1}} = S_j(\mathbf{F}_{S_j}), \quad \mathbf{F}_{S_1} = f_1(\mathbf{I}_{fn}), \tag{7}$$

where $S_j$ denotes the $j_{th}$ SRG, $1 \leq j \leq t-1$. The components of SRG are depicted in Fig. 3 (b). We define the input feature of the $j_{th}$ SRG as $\mathbf{F}_{S_j} \in \mathbb{R}^{H \times W \times C}$ and its channel as $C$. $\mathbf{F}_{S_j}$ continuously undergoes a *conv* layer, $n$ MABs, and a *conv* layer to add with an identity mapping:

$$\mathbf{F}_{S_{j+1}} = \mathbf{F}_{S_j} + M_n^j(\mathbf{F}_{M_n^j}), \quad \mathbf{F}_{M_{k+1}^j} = M_k^j(\mathbf{F}_{M_k^j}), \quad \mathbf{F}_{M_1^j} = \mathbf{F}_{S_j}, \tag{8}$$

where $M_k^j$ denotes the $k_{th}$ MAB of the $j_{th}$ SRG, $1 \leq k \leq n-1$. MAB is the basic building block and the most significant component of SMNet. The details of MAB are depicted in Fig. 3 (c). We customize MAB with the following motivations: **(i)** Multi-scale feature fusion can increase the receptive field and multi-resolution contextual information can cover rich auto-correlation, which provides more sufficient spatial representations for noise fitting. **(ii)** The noise level decreases as the scale increases and nonlinear sampling operations can increase the richness of the mapping in the potential space of real noise. Therefore, we exploit parallel multi-resolution branch aggregation from top to bottom and bottom to top to facilitate the learning of complex real noise. **(iii)** Specifically, during the feature downsampling, general downsample operation damages the image information, resulting in pixel discontinuity and jagged artifact. To alleviate these issues, we exploit Shift-Invariant Downsample [46] that copes with the discontinuity by using continuous pooling and

| Methods | SIDD [47] PSNR ↑ | SSIM ↑ | DND [48] PSNR ↑ | SSIM ↑ | Methods | PolyU [40] PSNR ↑ | SSIM ↑ | Nam [49] PSNR ↑ | SSIM ↑ |
|---|---|---|---|---|---|---|---|---|---|
| DnCNN-B [10] | 23.66 | 0.583 | 32.43 | 0.790 | RDN [12] | 37.94 | 0.946 | 38.16 | 0.956 |
| CBDNet [50] | 33.28 | 0.868 | 38.06 | 0.942 | FFDNet+ [19] | 38.17 | 0.951 | 38.81 | 0.957 |
| RIDNet [44] | 38.71 | 0.914 | 39.26 | 0.953 | TWSC [51] | 38.68 | 0.958 | 38.96 | 0.962 |
| AINDNet [52] | 39.15 | 0.955 | 39.53 | 0.956 | CBDNet [50] | 38.74 | 0.961 | 39.08 | 0.969 |
| VDN [53] | 39.23 | 0.955 | 39.38 | 0.952 | RIDNet [44] | 38.86 | 0.962 | 39.20 | 0.973 |
| CycleISP [22] | 39.52 | 0.957 | 39.56 | 0.956 | VDN [53] | 39.04 | 0.965 | 39.68 | 0.976 |
| MPRNet [54] | 39.71 | 0.958 | 39.80 | 0.954 | MPRNet [54] | 39.07 | 0.969 | 39.41 | 0.974 |
| MIRNet [55] | 39.72 | 0.959 | 39.88 | 0.956 | MIRNet [55] | 39.18 | 0.973 | 39.57 | 0.979 |
| **RIDNet* (Ours)** | 39.25 | 0.956 | 39.55 | 0.955 | **RIDNet* (Ours)** | 39.54 | 0.971 | 39.69 | 0.975 |
| **MPRNet* (Ours)** | 40.06 | 0.960 | 40.18 | 0.961 | **MPRNet* (Ours)** | 40.48 | 0.982 | 40.72 | 0.984 |
| **MIRNet* (Ours)** | 40.07 | 0.960 | 40.25 | 0.962 | **MIRNet* (Ours)** | 40.55 | 0.983 | 40.78 | 0.986 |

Table 1: Comparison on four benchmarks. * denotes denoisers finetuned with images generated by PNGAN.

filtering operation, preserving rich cross-correlation information between original and downsampled images. **(iv)** To efficiently capture continuous channel correlation and avoid information loss, we use the 1D channel attention module, Fast Channel Attention (FCA) instead of the general 2D convolution attention module. The input feature, $\mathbf{F}_{M_k^j} \in \mathbb{R}^{H \times W \times C}$ is fed into three parallel multi-scale paths:

$$\mathbf{F}^1_{M_k^j} = FCA(\mathbf{F}_{M_k^j}), \quad \mathbf{F}^2_{M_k^j} = f^2_{up}(FCA(f^2_{sid}(\mathbf{F}_{M_k^j}))), \quad \mathbf{F}^4_{M_k^j} = f^4_{up}(FCA(f^4_{sid}(\mathbf{F}_{M_k^j}))), \quad (9)$$

where $FCA$ denotes Fast Channel Attention. $f^2_{up}$ denotes a *conv* layer after bilinear interpolation up-sampling, 2 is the scale factor. $f^4_{up}$ is similarly defined. $f^2_{sid}$ means Shift-Invariant Downsample [46], 2 is also the scale factor. $f^4_{sid}$ is similarly defined. Subsequently, the output feature is derived by:

$$M_k^j(\mathbf{F}_{M_k^j}) = \mathbf{F}_{M_k^j} + f([\mathbf{F}^1_{M_k^j}, \mathbf{F}^2_{M_k^j}, \mathbf{F}^4_{M_k^j}]), \quad (10)$$

where $f$ represents the last *conv* layer, $[\cdot, \cdot, \cdot]$ denotes the concatenating operation. The architecture of FCA is shown in Fig. 3 (d). We define the input feature as $\mathbf{F}_d$, then FCA can be formulated as:

$$FCA(\mathbf{F}_d) = \mathbf{F}_d \cdot \left(1 + \sigma\big(f_{1DC}(GAP(\mathbf{F}_d))\big)\right), \quad (11)$$

where $\sigma$ represents the Sigmoid activation function, $GAP$ means global average pooling along the spatial wise, $f_{1DC}$ denotes 1-Dimension Convolution. In this work, we set $t = 3$, $n = 2$, and $C = 64$.

### 2.4 Overall Training Objective

In addition to the aforementioned losses, we employ a perceptual loss function that assesses a solution with respect to perceptually relevant characteristics (e.g., the structural contents and detailed textures):

$$\mathcal{L}_p = \left|\left|VGG(\mathbf{I}_{fd}) - VGG(\mathbf{I}_{rd})\right|\right|_2^2, \quad \mathbf{I}_{fd} = D_d(\mathbf{I}_{fn}), \quad \mathbf{I}_{rd} = D_d(\mathbf{I}_{rn}), \quad (12)$$

where $VGG$ denotes the last feature map of VGG16 [56]. Eventually, the training objective is:

$$\mathcal{L} = \mathcal{L}_1 + \lambda_p \cdot \mathcal{L}_p + \lambda_{Ra} \cdot (\mathcal{L}_D + \mathcal{L}_G), \quad (13)$$

where $\lambda_p$ and $\lambda_{Ra}$ are two hyper-parameters controlling the importance balance. The proposed PNGAN framework is end-to-end trained by minimizing $\mathcal{L}$. Note that the parameters in $D_d$ and VGG16 are fixed. Each mini-batch training procedure is divided into two steps: (i) Fix $D$ and train $G$. (ii) Fix $G$ and train $D$. This pixel-level adversarial training scheme promotes $D$ the ability to distinguish fake noisy images from real noisy images and allows $G$ to learn to create the solutions that are highly similar to real camera noisy images and thus difficult to be classified by $D$.

## 3 Experiment

### 3.1 Experiment Setup

**Datasets.** We first use SIDD [47] `train` set to train $D_d$. Then we fix $D_d$ to train $G$ on the same set. Subsequently, $G$ uses clean images from DIV2K [57], Flickr2K [58], BSD68 [59], Kodak24 [60], and Urban100 [61] to generate realistic noisy-clean image pairs. We use the generated data and SIDD `train` set jointly to finetune real denoisers and evaluate them on four real denoising benchmarks: SIDD [47], DND [48], PolyU [40], and Nam [49]. The images in SIDD [47] are collected using five smartphone cameras in 10 static scenes. There are 320 image pairs for training and 1,280 image patch pairs for validation. DND [48] composes 50 noisy-clean image pairs captured by 4 consumer

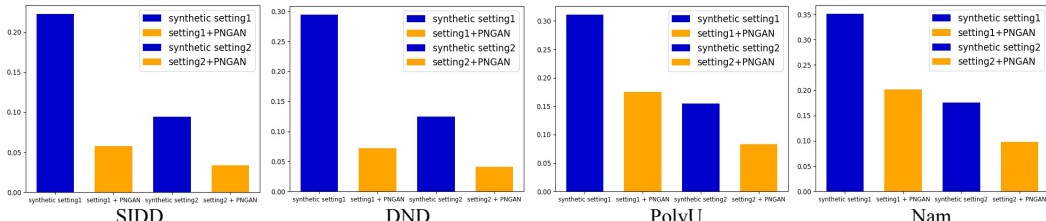

| | SIDD | DND | PolyU | Nam |

Figure 4: Domain discrepancy comparisons. We use the metric, Maximum Mean Discrepancy (MMD) to measure the domain discrepancy between synthetic and real noisy datasets, PNGAN generating and real noisy datasets. Under both setting1 and 2, the discrepancy decreases significantly when PNGAN is applied.

| Methods | SIDD [47] | | | | | DF2K [57, 58] | | | |
| | S1 | S1 + PNGAN | S2 | S2+PNGAN | Real | S1 | S1 + PNGAN | S2 | S2+PNGAN |
|---|---|---|---|---|---|---|---|---|---|
| RIDNet | 22.55 | **37.92** (+15.37) | 36.13 | **38.71** (+2.58) | 38.69 | 22.55 | **32.10** (+9.55) | 33.98 | **38.14** (+4.16) |
| MPRNet | 22.86 | **38.52** (+15.66) | 36.52 | **39.53** (+3.01) | 39.45 | 22.85 | **32.82** (+9.97) | 34.19 | **38.61** (+4.42) |
| MIRNet | 22.83 | **38.76** (+15.93) | 36.55 | **39.57** (+3.02) | 39.58 | 23.08 | **32.34** (+9.26) | 34.26 | **38.72** (+4.46) |

Table 2: Training denoisers with different data from scratch. PSNR is reported. S1,2 = synthetic setting1,2.

cameras. 1,000 patches at size 512×512 are cropped from the collected images. PolyU [40] consists of 40 real camera noisy images. Nam [49] is composed of real noisy images of 11 static scenes.

**Implementation Details.** We set the hyper-parameter $\lambda_p = 6 \times 10^{-3}$, $\lambda_{Ra} = 8 \times 10^{-4}$. For synthetic setting1, we set the noise intensity, $\sigma_n = 50$. For synthetic setting2, we directly exploit CycleISP to generate the synthetic noisy input. All the sub-modules ($D_d$, $G$, and $D$) are trained with the Adam [62] optimizer ($\beta_1 = 0.9$ and $\beta_1 = 0.9999$) for $7 \times 10^5$ iterations. The initial learning rate is set to $2 \times 10^{-4}$. The cosine annealing strategy [63] is employed to steadily decrease the learning rate from the initial value to $10^{-6}$ during the training procedure. Patches at size 128×128 cropped from training images are fed into the models. The batch size is set as 8. The horizontal and vertical flips are performed for data augmentation. All the models are trained on RTX8000 GPUs. In the finetuning phase, the learning rate is set to $1 \times 10^{-6}$, other settings remain unchanged.

### 3.2 Quantitative Results

**Domain Discrepancy Validation.** We use the widely applied metric, Maximum Mean Discrepancy (MMD) [64] to measure the domain discrepancy between synthetic and real-world noisy images, PNGAN generating, and real noisy images on four real noisy benchmarks. For DND, we derive a pseudo clean version by denoising the real noisy counterparts with a pre-trained MIRNet [55]. Then we use the pseudo clean version to synthesize noisy images. The results are depicted as a histogram in Fig. 4. For setting1, the domain discrepancy decreases by 74%, 75%, 44%, and 43% on SIDD, DND, PolyU, and Nam when PNGAN is exploited. For setting2, the discrepancy decreases by 64%, 67%, 46%, and 44%. These results demonstrate that PNGAN can narrow the discrepancy between synthetic and real noisy datasets. Please refer to the supplementary for detailed calculation process.

**Comparison with SOTA Methods.** We use the generated noisy-clean image pairs (setting2) to finetune a series of denoisers. We compare our models with SOTA algorithms on four real denoising datasets: SIDD, DND, PolyU, and Nam. The results are reported in Tab. 1. ✱ denotes denoisers finetuned with image pairs generated by PNGAN. We have the following observations: **(i)** Our denoisers outperform SOTA methods by a large margin. Specifically, MPRNet* and MIRNet* exceed the recent best method MIRNet by 0.34 and 0.35 dB on SIDD, 0.30 and 0.37 dB on DND. RIDNet*, MPRNet*, and MIRNet* surpass the best performers by 0.36, 1.30, and 1.37 dB on PolyU and 0.01, 1.04, and 1.10 dB on Nam. **(ii)** Compared with the counterparts that are not finetuned, our models achieve a significant promotion. In particular, RIDNet* is 0.54, 0.29, 0.68, and 0.49 dB higher than RIDNet on SIDD, DND, PolyU, and Nam. MPRNet* achieves 0.35, 0.38, 1.41, and 1.31 dB gain than MPRNet on SIDD, DND, PolyU, and Nam. MIRNet* is improved by 0.35, 0.37, 1.37, and 1.21 dB. This evidence clearly suggests the high similarity between PNGAN generating and real noisy images. Denoisers adapted with our fake image pairs generalize better across different benchmarks.

**Train from Scratch.** For more strong comparisons, we use the fake noisy images generated from clean SIDD `train` and DF2K (DIV2K+Flicker2K) respectively to train denoisers from scratch. The PSNR results evaluated on SIDD `test` are listed in Tab. 2. All models are trained with the same experiment schedule except the training data. It can be observed: **(i)** On SIDD `train`, when PNGAN is applied to setting1, denoisers are promoted by ∼ 15.65 dB and only ∼ 0.84 dB lower than those

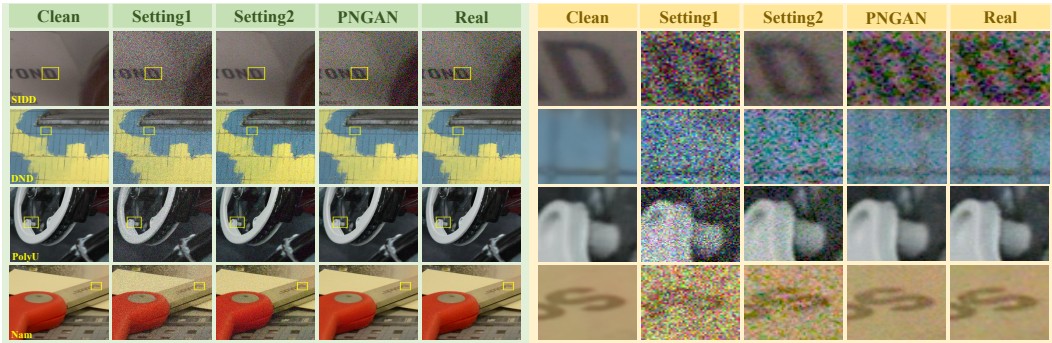

Figure 5: Visual comparisons of noisy images on SIDD, DND, PolyU, and Nam. Please zoom in.

| Methods | PNGAN Component | | | | Generator Architecture | | | |
| --- | --- | --- | --- | --- | --- | --- | --- | --- |
| | Baseline1 | + $D_d$ | + $D$ | + $\mathcal{L}_p$ | Baseline2 | + Multi-scale | + SID | + FCA |
| RIDNet | 14.54 | 35.37 (+20.83) | 37.49 (+2.12) | 37.92 (+0.43) | 35.62 | 37.01 (+1.39) | 37.23 (+0.22) | 37.92 (+0.69) |
| MPRNet | 14.25 | 36.26 (+22.01) | 38.27 (+2.01) | 38.52 (+0.25) | 36.28 | 37.47 (+1.19) | 37.86 (+0.39) | 38.52 (+0.66) |
| MIRNet | 13.57 | 36.15 (+22.58) | 38.28 (+2.13) | 38.76 (+0.48) | 36.39 | 37.66 (+1.27) | 37.89 (+0.23) | 38.76 (+0.87) |

Table 3: Ablation study of PNGAN component and the noise generator architecture. PSNR is reported.

trained with real data (SIDD `train` set). While applying PNGAN to setting2 (CycleISP), denoisers are improved by $\sim 2.87$ dB. Surprisingly, in this case, denoisers achieve almost the same performance as those trained with real data. The relative error is 0.2%. **(ii)** To validate the generality of PNGAN, we also adopt synthetic DF2K noisy-clean image pairs to train denoisers. As shown in the right part of Tab. 2, when PNGAN is applied to setting1, denoisers are promoted by $\sim 9.59$ dB. While applying PNGAN to setting2, denoisers are improved by $\sim 4.35$ dB and only $\sim 0.75$ dB lower than those trained with SIDD real `train` set. These results convincingly demonstrate: **(i)** The generated noise is highly similar to the real noise especially when PNGAN is applied to synthetic setting2. **(ii)** PNGAN can significantly narrow the domain discrepancy between synthetic and real-world noise.

### 3.3 Qualitative Results

**Visual Examinations of Noisy Images.** To intuitively evaluate the generated noisy images, we provide visual comparisons of noisy images on the four real noisy datasets, as shown in Fig. 5. Note that the clean image of DND is pseudo, denoised from its noisy version by a MIRNet. The left part depicts noisy images from SIDD, DND, PolyU, and Nam (top to down). The right part exhibits the patches cropped by the yellow bboxes, from left to right: clean, synthetic setting1, setting2 (CycleISP), PNGAN generating, and real noisy images. As can be seen from the zoom-in patches: **(i)** Noisy images synthesized by setting1 is signal-independent. The distribution and intensity remain unchanged across diverse scenes, indicating the characteristics of AWGN fundamentally differ from those of the real noise. **(ii)** Noisy images generated by PNGAN are closer to the real noise than those synthesized by setting2 visually. Noise synthesized by setting2 shows randomness that is obviously inconsistent with the real noise in terms of intensity and distribution. While PNGAN can model spatio-chromatically correlated and non-Gaussian noise more accurately. **(iii)** Even if passing through the same camera pipeline, different shooting conditions lead to the diversity of real noise. It's unreasonable for the noise synthesized by CycleISP to show nearly uniform fitting to different input images. In contrast, PNGAN can adaptively simulate more sophisticated and photo-realistic models. This adaptability allows PNGAN to show robust performance across different real noisy datasets.

**Visual Comparison of Denoised Images.** We compare the visual results of denoisers before and after being finetuned (denoted with *) with the generated data in Fig. 4. We observe that models finetuned with the generated data are more effective in real noise removal. Furthermore, they are capable of preserving the structural content, textural details, and spatial smoothness of the homogeneous regions. In contrast, original models either yield over-smooth images sacrificing fine textural details and structural content or introduce redundant blotchy texture and chroma artifacts.

### 3.4 Ablation Study

**Break-down Ablations.** We perform break-down ablations to evaluate the effects of PNGAN components and SMNet architecture. We select setting1 to synthesize the noisy input from SIDD `train` set. Then we use the generated data only to train the denoisers from scratch and evaluate them on SIDD `test`. The PSNR results are reported in Tab. 3. **(i) Firstly**, $G$ is set as SMNet to validate the effects of PNGAN components. We start from Baseline1, no discriminator is used and the $\mathcal{L}_1$ loss is

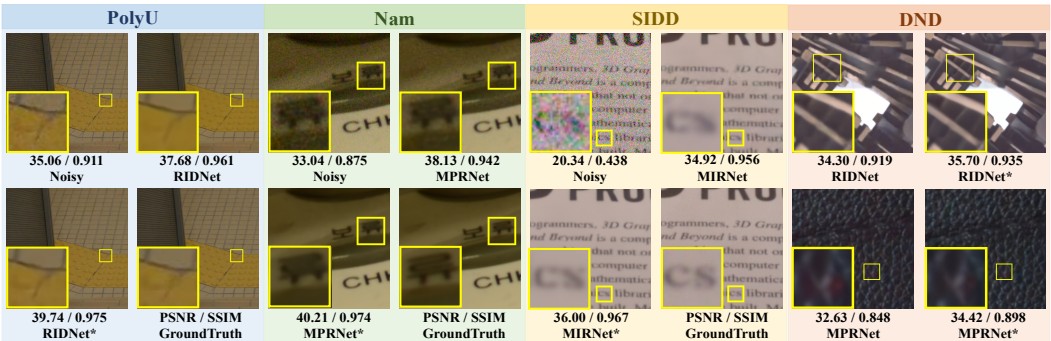

Figure 6: Visual results of denoisers before and after being finetuned with fake data. Please zoom in.

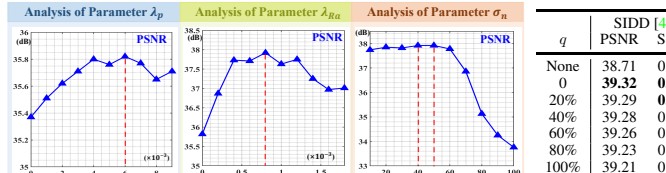

Figure 7: Parameter analysis of $\lambda_p$, $\lambda_{Ra}$, and $\sigma_n$.

| $q$ | SIDD [47] PSNR | SIDD [47] SSIM | PolyU [40] PSNR | PolyU [40] SSIM | Nam [49] PSNR | Nam [49] SSIM | Total PSNR | Total SSIM |
|---|---|---|---|---|---|---|---|---|
| None | 38.71 | 0.914 | 38.86 | 0.962 | 39.20 | 0.973 | 38.76 | 0.929 |
| 0 | **39.32** | **0.957** | 38.01 | 0.949 | 38.34 | 0.958 | 38.92 | 0.955 |
| 20% | 39.29 | **0.957** | 38.45 | 0.959 | 38.87 | 0.970 | 39.03 | 0.958 |
| 40% | 39.28 | 0.956 | 39.02 | 0.966 | 39.26 | 0.973 | 39.20 | 0.959 |
| 60% | 39.26 | 0.956 | 39.54 | 0.971 | 39.69 | 0.975 | **39.35** | **0.961** |
| 80% | 39.23 | 0.955 | 39.56 | 0.972 | 39.72 | **0.976** | 39.33 | 0.960 |
| 100% | 39.21 | 0.955 | **39.57** | **0.972** | **39.73** | **0.976** | 39.33 | 0.960 |

Table 4: Analysis of the finetuning data ratio $q$.

directly performed between $\mathbf{I}_{fn}$ and $\mathbf{I}_{rn}$ in Eq. (2). Denoisers trained with the generated data collapse dramatically, implying the naive strategy mentioned in Sec. 2.2 is unfeasible. When $D_d$ is applied, the denoisers are promoted by 21.81 dB on average. **In addition**, the PSNR and SSIM between the denoised counterparts of generated and real noisy images are 39.14 dB and 0.928 on average respectively. This evidence indicates that $D_d$ successfully conducts the image domain alignment as mentioned in Sec. 2.2. **Subsequently**, we use an image-level $D$ with stride *conv* layers to classify whether the whole generated image is real. Nonetheless, the performance of denoisers remains almost unchanged. After deploying $D$, the models are improved by $\sim$2.09 dB, suggesting that the pixel-level noise model is more in line with real noise scenes and benefits generating more realistic noisy images. When $\mathcal{L}_p$ is used, the denoisers gain a slight improvement by about 0.39 dB, indicating $\mathcal{L}_p$ facilitates yielding more vivid results. **(ii) Secondly**, we only change the architecture of $G$ to study the effects of its components. We start from Baseline2 that doesn't exploit multi-scale feature fusion, SID, and FCA. When we add two different scale branches and use bilinear interpolation to downsample and upsample, denoisers trained with the generated images are promoted by about 1.28 dB. After applying SID and FCA, the denoisers further gain 0.28 and 0.74 dB improvement on average. These results convincingly demonstrate the superiority of the proposed SMNet in real-world noise fitting.

**Parameter Analysis.** We adopt RIDNet as the baseline to perform parameter analysis. We **firstly** validate the effects of $\lambda_p$, $\lambda_{Ra}$ in Eq. (13), and the noise intensity of setting1, i.e., $\sigma_n$. We change the parameters, train $G$, use $G$ to generate realistic noisy images from clean images of SIDD train set, train RIDNet with the generated data, and evaluate its performance on SIDD test set. When analyzing one parameter, we fix the others at their optimal values. The PSNR results are shown in Fig. 7. The optimal setting is $\lambda_p = 6 \times 10^{-3}$, $\lambda_{Ra} = 8 \times 10^{-4}$, and $\sigma_n = 40$ or 50. **Secondly**, we evaluate the effect of the ratio of finetuning data. We denote the ratio of extended training data (setting2) to SIDD real noisy training data as $q$. We change the value of $q$, finetuned the original RIDNet, and test on three real denoising datasets: SIDD, PolyU, and Nam. The results are listed in Tab. 4. When $q = 0$, all the finetuning data comes from SIDD train set, RIDNet achieves the best performance on SIDD. However, its performance on PolyU and Nam degrades drastically due to the domain discrepancy between different real noisy datasets. We gradually increase the value of $q$ to study its effects. The average performance on the three datasets yields the maximum when $q = 60\%$.

## 4 Conclusion

Too much research focuses on designing a CNN architecture for real noise removal. In contrast, this work investigates how to generate more realistic noisy images so as to boom the denoising performance. We first formulate a noise model that treats each noisy pixel as a random variable. Then we propose a novel framework PNGAN to perform the image and noise domain alignment. For better noise fitting, we customize an efficient architecture, SMNet as the generator. Experiments show that noise generated by PNGAN is highly similar to real noise in terms of intensity and distribution. Denoisers finetuned with the generated data outperform SOTA methods on real denoising datasets.

## Acknowledgement

This work is jointly supported by the NSFC fund (61831014), in part by the Shenzhen Science and Technology Project under Grant (ZDYBH201900000002, JCYJ20180508152042002, CJGJZD20200617102601004).

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
