# The Supplementary Materials of the Main Paper: Learning to Generate Realistic Noisy Images via Pixel-level Noise-aware Adversarial Training

**Yuanhao Cai** [1,2], **Xiaowan Hu** [1,2], **Haoqian Wang** [1,2,*],
**Yulun Zhang** [3], **Hanspeter Pfister** [4], **Donglai Wei** [5]
[1] Shenzhen International Graduate School, Tsinghua University,
[2] Shenzhen Institute of Future Media Technology,
[3] ETH Zürich, [4] Harvard University, [5] Boston College

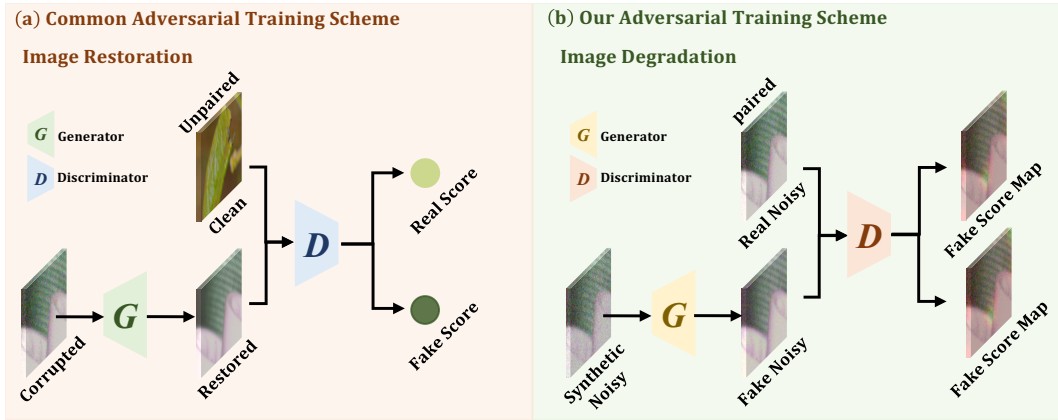

Figure 1: Comparison between the widely used adversarial training scheme in image restoration and the PNGAN framework. (a) GAN is usually adopted as a semi-supervised strategy for image restoration. (b) In contrast, our PNGAN is proposed for image degradation learning in real-world noisy scenes.

In this document, we share more discussions, details, quantitative and qualitative results, broader impacts, limitations, and future works not included in our main paper because of the space constraints.

## 1 Discussion with previous works

Image degradation and restoration are two important research fields in computational photography. However, too much work is dedicated to handling the image restoration problem and the image degradation remains under-studied. In contrast, this work essentially belongs to the field of image degradation and focuses on implicitly learning how corruption is created in real-world noisy scenes. This is the most significant difference between the PNGAN and previous image restoration works.

## 2 Discussion with other GANs in Low-level Vision

As depicted in Fig. 1, we compare the common adversarial training scheme in image restoration and our PNGAN framework. **(i)** In the fields of image restoration, GAN is usually adopted as a semi-supervised strategy. Specifically, as shown in Fig. 1, the restorer that maps the degraded image into a nearly corruption-free version is adopted as the generator. Then the restored image concatenated with another unpaired clean image is fed into a two-category classifier (discriminator).

*Haoqian Wang is the corresponding author, email: wanghaoqian@tsinghua.edu.cn

35th Conference on Neural Information Processing Systems (NeurIPS 2021).

Then the discriminator outputs a score for each input image indicating how realistic the input image is. With this adversarial training scheme, a large amount of unpaired data can be utilized to promote the performance and generality of the restorer. Simultaneously, this semi-supervised mechanism also facilitates yielding a perceptually pleasing restored image. However, this kind of method is customized to handle the image restoration problem. **(ii)** In contrast, our PNGAN is proposed for image degradation learning in real-world noisy scenes. In particular, during the training phase, a synthetic noisy image that contains a random distribution is fed into the generator to create the fake noisy image. Subsequently, the paired real and fake noisy images are fed into the discriminator. Then the discriminator outputs a score map for each input image of $D$. The position on a score map indicates how realistic the corresponding pixel of the input image is.

## 3 Details of MMD Calculation

In Sec. 3.2 - **Domain Discrepancy Validation** (L 254-255) of the main paper, we use the widely applied metric, Maximum Mean Discrepancy (MMD [1]) to measure the domain discrepancy between synthetic and real-world noisy images, PNGAN generating, and real noisy images on four real noisy benchmarks. Here, we provide more details.

### 3.1 Motivation of Using MMD

Due to the lack of Image Quality Assessment (IQA) for corrupted images, it's hard to directly evaluate the quality of the generated noisy images. Thus, we treat fake and real noisy data as two distributions and adopt the widely applied metric, MMD in transfer learning to measure the discrepancy.

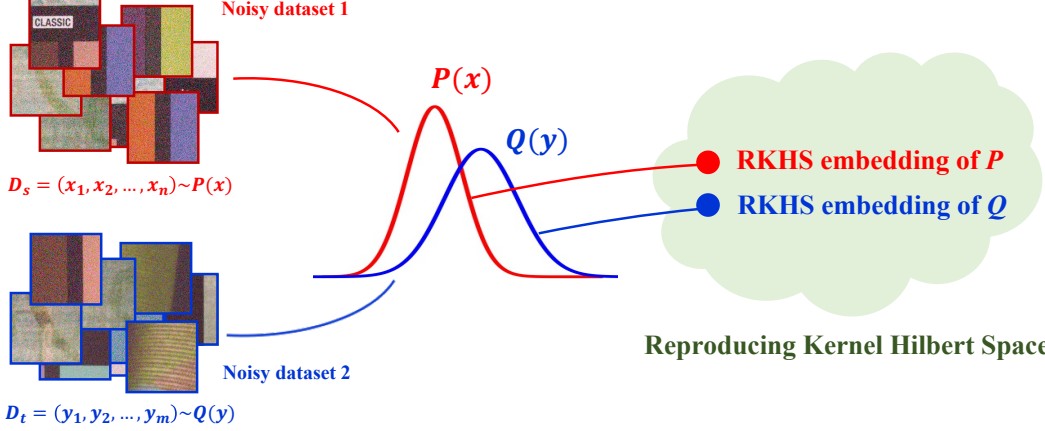

Figure 2: Illustration of using MMD to calculate the domain discrepancy of different noisy image datasets.

### 3.2 Definition of MMD

We define $\mathcal{X}$ as a non-empty collection, $k : \mathcal{X} \times \mathcal{X} \mapsto \mathbb{R}$ as a positive definite kernel fucntion and $\phi : \mathcal{X} \mapsto \mathcal{H}$ as the feature space mapping. The Reproducing Kernel Hilbert Space (RKHS) of $\mathcal{X}$ is denoted as $\mathcal{H}$. The illustration of using MMD to calculate the domain discrepancy of different nosiy image datasets is shown in Fig. 2. Given two noisy image datasets, $D_s = (x_1, x_2, ..., x_n) \sim P(x)$ and $D_t = (y_1, y_2, ..., y_m) \sim Q(y)$. The MMD between $D_s$ and $D_t$ is defined as:

$$\text{MMD}(P, Q) = \big|\big| \mathbb{E}_P[\phi(x)] - \mathbb{E}_Q[\phi(y)] \big|\big|_2 , \tag{1}$$

where $\mathbb{E}$ denotes taking the average value. However, it's hard to completely and accurately estimate the distributions of $P$ and $Q$. Thus, in practical application, the approximate value of Eq. (1) is calculated as follows:

$$\widehat{\text{MMD}}(P, Q) = \Big|\Big| \frac{1}{n} \sum_{i=1}^{n} \phi(x_i) - \frac{1}{m} \sum_{i=1}^{m} \phi(y_i) \Big|\Big|_2 . \tag{2}$$

Squares on both sides of Eq. (2):

$$\widehat{\mathrm{MMD}}(P,Q)^2 = \left|\left| \frac{1}{n}\sum_{i=1}^{n}\phi(x_i) - \frac{1}{m}\sum_{i=1}^{m}\phi(y_i) \right|\right|_2^2$$

$$= \left|\left| \frac{1}{n}\sum_{i=1}^{n}\phi(x_i) \right|\right|_2^2 + \left|\left| \frac{1}{m}\sum_{i=1}^{m}\phi(y_i) \right|\right|_2^2 - 2\left|\left| \frac{1}{n}\sum_{i=1}^{n}\phi(x_i)\frac{1}{m}\sum_{i=1}^{m}\phi(y_i) \right|\right|_2, \tag{3}$$

where

$$\left|\left| \frac{1}{n}\sum_{i=1}^{n}\phi(x_i) \right|\right|_2^2 = \frac{1}{n^2}(\phi(x_1)+\phi(x_2)+...+\phi(x_n))^T(\phi(x_1)+\phi(x_2)+...+\phi(x_n))$$

$$= \frac{1}{n^2}(\phi(x_1)^T\phi(x_1)+...+\phi(x_1)^T\phi(x_n)+$$
$$\phi(x_2)^T\phi(x_1)+...+\phi(x_2)^T\phi(x_n)+$$
$$...+\phi(x_n)^T\phi(x_1)+...+\phi(x_n)^T\phi(x_n))$$
$$= \frac{1}{n^2}(k(x_1,x_1)+k(x_1,x_2)+...+k(x_1,x_n)+$$
$$k(x_2,x_1)+k(x_2,x_2)+...+k(x_2,x_n)+$$
$$...+k(x_n,x_1)+k(x_n,x_2)+...+k(x_n,x_n)),$$
$$= \frac{1}{n^2}\sum_{i,j}k(x_i,x_j). \tag{4}$$

The same can be obtained:

$$\left|\left| \frac{1}{m}\sum_{i=1}^{m}\phi(y_i) \right|\right|_2^2 = \frac{1}{m^2}\sum_{i,j}k(y_i,y_j),$$

$$\left|\left| \frac{1}{n}\sum_{i=1}^{n}\phi(x_i)\frac{1}{m}\sum_{i=1}^{m}\phi(y_i) \right|\right|_2 = \frac{1}{mn}\sum_{i,j}k(x_i,y_j). \tag{5}$$

Eventually, Eq. (2) is formulated into:

$$\widehat{\mathrm{MMD}}(P,Q) = (\frac{1}{n^2}\sum_{i,j}k(x_i,x_j) + \frac{1}{m^2}\sum_{i,j}k(y_i,y_j) - \frac{2}{mn}\sum_{i,j}k(x_i,y_j))^{\frac{1}{2}}. \tag{6}$$

We adopt the widely used Gaussian kernel function as $k$. 100 images are randomly selected from two different noisy image datasets respectively. Then we use Eq. (6) to calculate the MMD.

## 4 More Qualitative Results

### 4.1 Visual Examinations of Noisy Images

**Real Noisy Datasets.** As shown in Fig. 3, we firstly provide more visual comparisons of noisy images on four real noisy datasets: PolyU [2], Nam [3], SIDD [4], and DND [5]. As can be seen from the zoom-in patches: **(i)** The synthetic noisy images of setting1 is signal-independent. The distribution and intensity of synthetic noise remain unchanged across diverse scenes. There is a significant distribution discrepancy between synthetic and real noise. In the real world, illumination changes and camera movement inevitably lead to spatial pixel misalignment and color or brightness deviation, which makes real camera noise fundamentally differ from the signal-independent synthetic noise. **(ii)** Noisy images generated by PNGAN are closer to the real noise than those synthesized by setting2 visually. Noise synthesized by setting2 shows randomness that is obviously inconsistent with the real noise in terms of intensity and distribution. While PNGAN can model spatio-chromatically correlated and non-Gaussian noise more accurately. **(iii)** Even if passing through the same camera pipeline, different shooting conditions lead to the diversity of real noise. It's unreasonable for the noise synthesized by CycleISP to show nearly uniform fitting to different input images. In contrast,

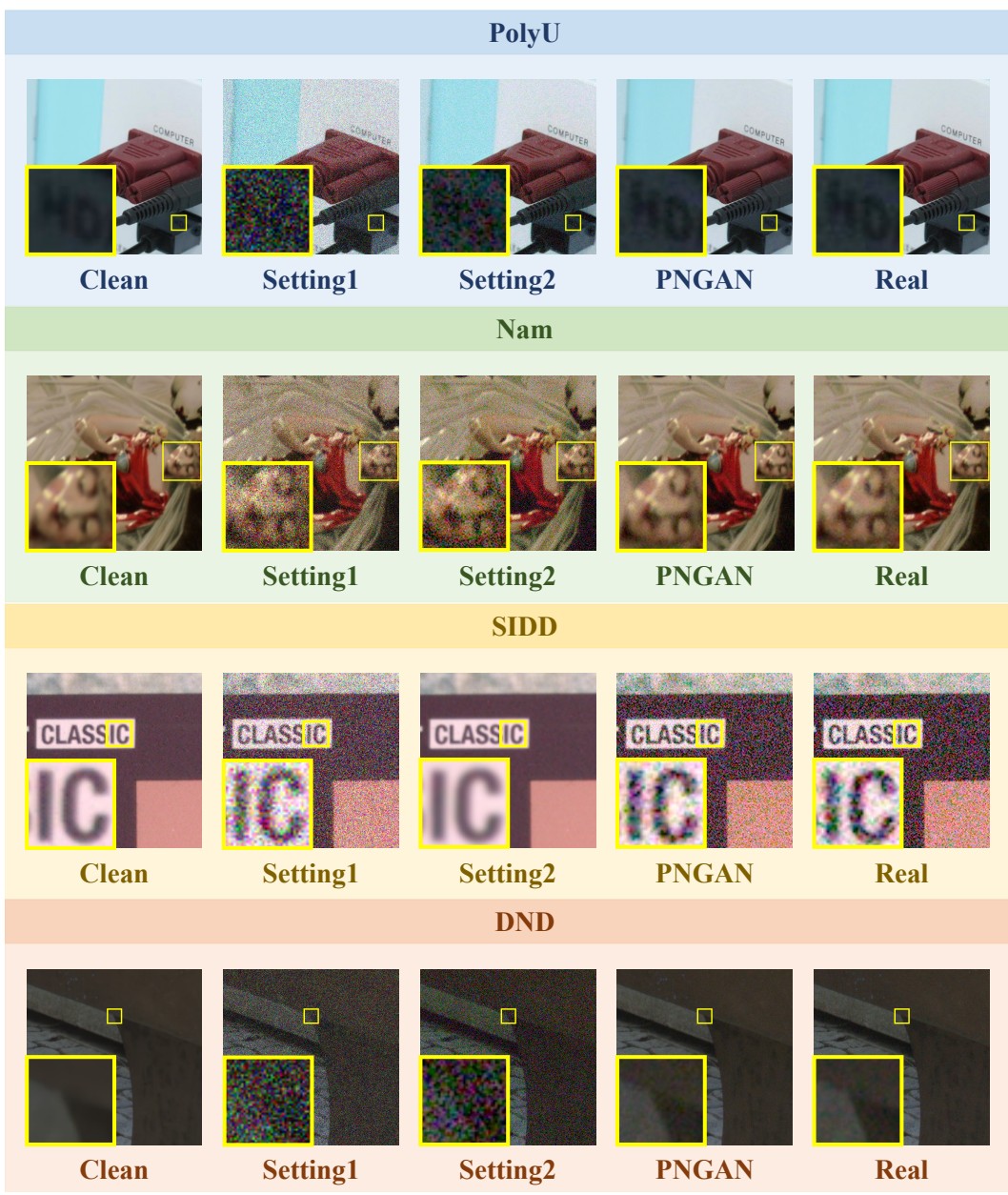

Figure 3: Visual comparisons of noisy images on four real noisy datasets: SIDD, DND, PolyU, and Nam.

PNGAN can adaptively simulate more sophisticated and photo-realistic models. This adaptability allows PNGAN to show promising performance across different real noisy datasets.

**HD Clean Datasets.** As shown in Fig. 4, we provide visual examinations of generated noisy images from HD clean datasets: BSD68 [6], Urban100 [7], DIV2K [8], Flickr2K [9], and Kodak24 [10]. These datasets are widely used to synthesize Gaussian noisy images and lack real noisy counterparts. Thus, we only compare noisy images synthesized with setting1 and noisy images generated by PNGAN. It can be observed from the zoom-in patches: **(i)** PNGAN is capable of yielding perceptually natural images and preserving the structural content and textural details of the original images. In contrast, synthetic setting1 fails in simulating the spatially variant noise and characterizing complex image textures. **(ii)** The PNGAN generating noise adaptively changes with scenes, illuminations, and shooting conditions while the synthetic noise is signal-independent and visually unpleasant. These comparisons clearly suggest the promising performance of PNGAN in realistic noise generating.

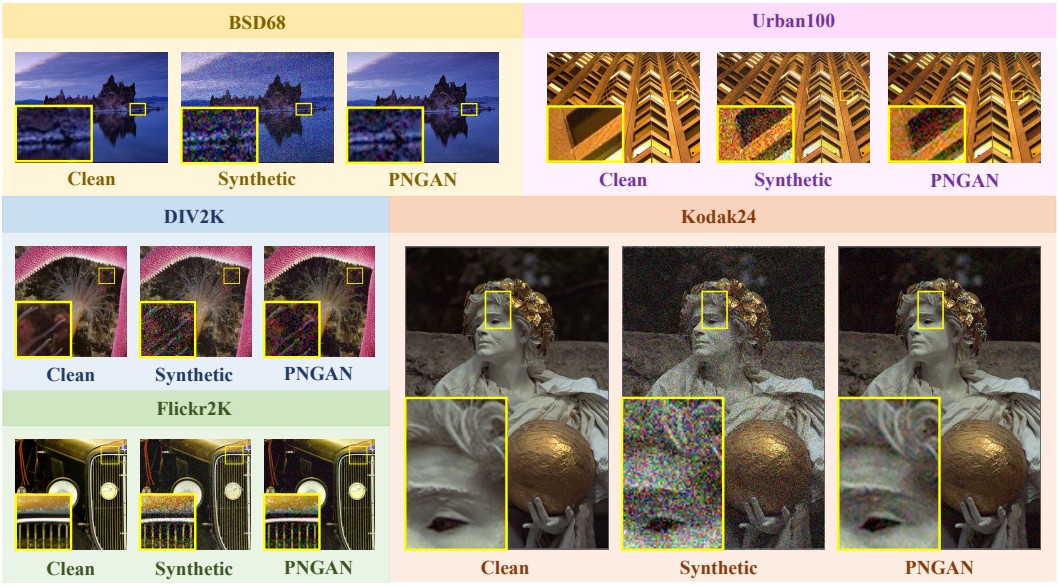

Figure 4: Visual comparisons of noisy images on BSD68, Urban100, DIV2K, Flickr2K, and Kodak24.

## 4.2 Visual Comparisons of Denoised Images

Fig. 5 depicts more visual examinations of denoised images. We compare the performance of denoisers before and after being finetuned with the fake data (denoted with *). It's observed that the finetuned denoisers are more effective in real noise removal. Furthermore, they are capable of preserving the original RGB color, structural content, textural details, and the spatial smoothness of the homogeneous regions without introducing redundant artifacts. In contrast, original denoisers either yield over-smooth images sacrificing fine textural details and structural content, produce RGB color distortion, or introduce redundant blotchy texture and chroma artifacts. These results suggest the high similarity between the generated and real noisy images and the effectiveness of PNGAN.

## 4.3 Visual Comparisons with DANet

DANet [11] also learns to generate realistic noisy images. We provide visual comparisons of noisy images generated by our PNGAN and DANet. Please zoom in for a better view. As depicted in Fig. 6, DANet suffers from severe chessboard effect, and introduces some black spots and artifacts when creating noisy images. In contrast, our PNGAN are more effective to generate visually pleasant realistic noise images. The distribution of the generated noise is much closer to that of real noise.

## 5 Broader Impact

Image denoising is a conventional task in computational photography. It has been studied for decades. Image denoising is widely applied in mobile photo restoration, medical imaging, and low-light image and video enhancement. Besides, image denoising usually functions as a preprocessing technique for many high-level vision tasks, i.e., object detection, semantic segmentation, and human pose estimation. Nowadays billions of mobile phones are widely used all over the world. This indicates the majority of images come from smartphone cameras having a small sensor size but large resolution. As a result, the presence of noise in images is inevitable, especially in low-light shooting conditions. Hence, image denoising techniques handling this problem are worth studying. The proposed PNGAN framework targets at addressing the data-hungry problem in image denoising. It could serve as a data augmentation strategy and promote the performance of data-driven denoising algorithms.

Until now, image denoising techniques have no negative impact yet. The proposed PNGAN doesn't present any negative foreseeable societal consequence, either.

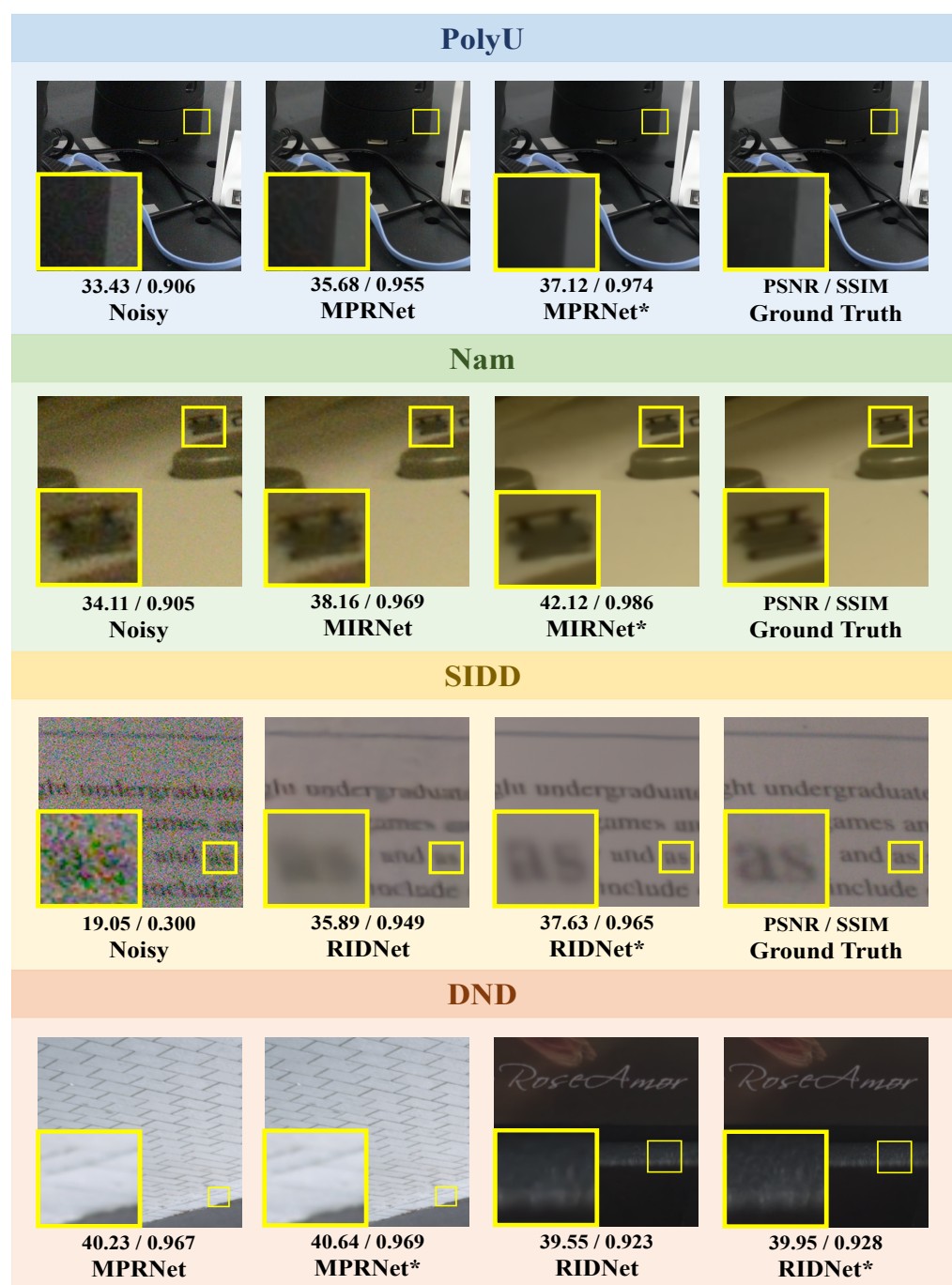

Figure 5: Visual results of denoisers before and after being finetuned with fake data. Please zoom in.

# 6 Limitation

The limitation of this work is that the quality of the generated noisy image depends on the input of the generator, i.e., the synthetic noisy image. More specifically, we take two examples. **(i) Firstly**, in Sec. 3.2 - **Quantitative Results** - **Train from scratch** (L 276-289) and Tab. 2 of the main paper, we respectively select synthetic setting1 and 2 to train $G$ and create fake noisy images. Then we use the generated data only to train denoisers. Denoisers trained with synthetic setting2 perform better than those trained with synthetic setting1. Denoisers trained with fake noisy images created

| Clean | DANet | PNGAN | Real |
|---|---|---|---|

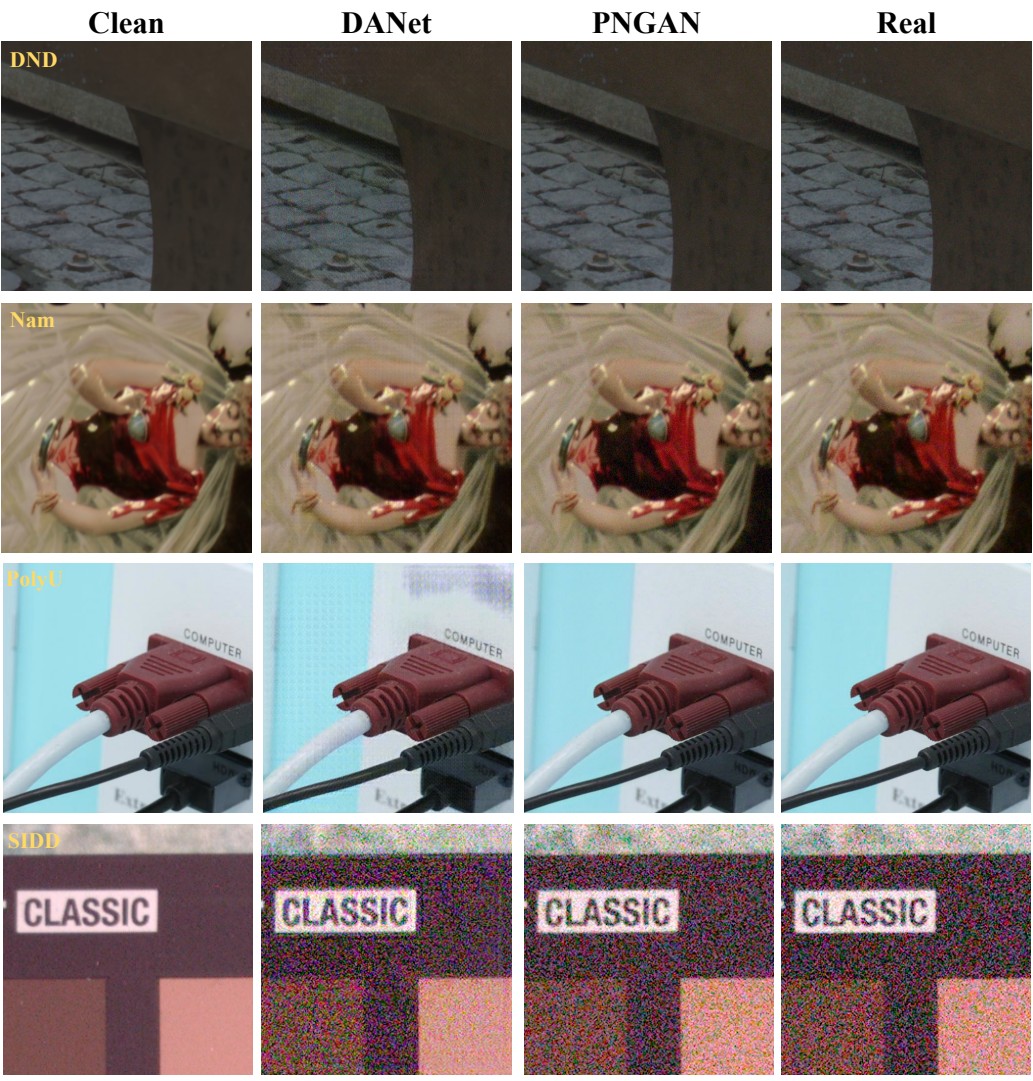

Figure 6: Visual comparisons of noisy images generated by the proposed PNGAN and DANet [11].

by PNGAN (setting2) perform better than those trained with fake noisy images created by PNGAN (setting1). These results indicate that the noise distribution synthesized with setting2 is closer to the real noise distribution than that synthesized with setting1. As a result, when we select synthetic setting2 as input, the generated noise distribution is more similar to the real noise distribution. **(ii) Secondly**, in Sec. 3.4 - **Ablation Study** - **Parameter Analysis** (L 335-340) and Fig. 7 of the main paper, we analyze the effect of the noise intensity of synthetic setting1, i.e., $\sigma_n$. when $\sigma_n$ is too high, the performance of denoisers trained with the generated data degrades drastically. This implies that when synthetic noisy images with high-intensity AGWN are selected as the input of $G$, it's hard for $G$ to model the real noise distribution from such initial value and create realistic noisy images.

However, this is a common limitation in the field of deep learning. The fundamental reason is that CNN-based methods are mainly data-driven. The performance changes as the training data changes.

## 7  Deep Understanding of Our Work

As shown in Fig. 7, Image degradation and restoration are two important research fields in low-level vision. With the development of deep CNN, image restoration has witnessed significant progress. Too much research of image restoration focuses on designing a CNN and training it under the supervision

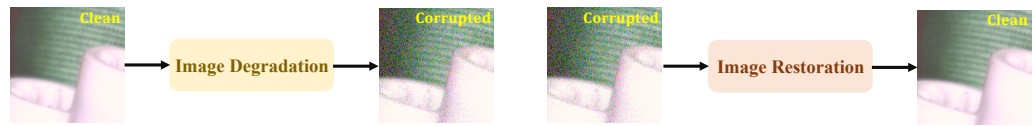

Figure 7: Image restoration aims to restore a clean image from its corrupted counterpart. Image degradation targets at generating a corrupted image from its clean version. The two processes reverse each other.

of a large-scale corrupted-clean image pair dataset to yield a better performance. Nonetheless, this research scheme is not suitable for image degradation. We analyze the reasons as follows: **(i) Firstly and most importantly**, image degradation is not a simple regression task because the real corrupted image is not a stable constant but a random variable. Specifically, the corruption on a real degraded image is sophisticated and changeable. The same scene will show quite different degradations under different imaging conditions. For instance, different weather conditions will lead to quite different rainy and foggy images of the same scene. This implies real degraded images should be treated as a random variable. Thus, naively formulating the image degradation problem into a simple regression task under the supervision of a large amount of real clean-corrupted image pair dataset may easily cause the non-convergence issue and fail in real degraded image generation. How to construct an effective supervision over real corrupted images still bothers researchers. **(ii) Secondly**, no universal Image Quality Assessment (IQA) has been established in the field of image degradation. This leads to such an embarrassing problem that it's hard to assess how realistic the created corrupted image is. The widely used PSNR and SSIM are two strict metrics in image restoration. A small pixel misalignment will easily cause a large change in the indicator. Therefore, these two metrics are not suitable for image degradation as the real-world corruption is much more sophisticated and changeable.

This work essentially belongs to the field of image degradation. **(i) Firstly**, Different from the previous research scheme of image restoration, we investigate the image degradation in the real noisy scene. Specifically, we treat each pixel of a real noisy image as a random variable and propose a PNGAN learning to model its distribution. **(ii) Secondly**, to assess the quality of the generated noisy images, we provide extensive visual examinations and directly utilize the MMD metric to evaluate the domain discrepancy between fake and real noisy images. Besides, we employ the proposed PNGAN to perform data augmentation for a series of real denoisers whose performances are then promoted by a large margin. The experimental results of training denoisers from scratch using only fake data also clearly suggest the high similarity between fake and real noisy images.

## 8 Future Work

The future works include: **(i)** Generalizing the PNGAN framework to other image degradation situations, e.g., rainy, foggy, blurry, and low-resolution scenes. **(ii)** Designing a better input setting to improve the quality of the generated fake image. **(iii)** Modifying the architecture of $G$ and $D$.

**General Corruption Modeling.** Specifically, for future work **(i)**, we generalize Eq. (1) of our main paper into a real corruption model as follow:

$$\mathbf{I}_{rc}[i] = \hat{\mathbf{I}}_{clean}[i] + \mathbf{C}[i], \quad R(\mathbf{I}_{rc})[i] = \hat{\mathbf{I}}_{clean}[i], \quad 1 \leq i \leq H \times W, \tag{7}$$

where $\mathbf{I}_{rc}[i]$ denotes the real corrupted image $\mathbf{I}_{rc} \in \mathbb{R}^{H \times W \times 3}$ at pixel $i$ and $\hat{\mathbf{I}}_{clean} \in \mathbb{R}^{H \times W \times 3}$ is the predicted clean counterpart of $\mathbf{I}_{rc}$, it's restored by the real restorer $R$. $C$ denotes the real corruption. Each $\mathbf{C}[i]$ is a random noise variable with an unknown probability distribution. Therefore, each $\mathbf{I}_{rc}[i]$ can also be viewed as a distribution-unknown random variable. We change the $D_d$ in the PNGAN framework by $R$, select different input settings according to the degradation situations, and propose to use the generalized version of the PNGAN framework to construct the image domain alignment and corruption domain alignment. This idea sounds reasonable and interesting. **However, we haven't done experiments to validate its effectiveness.** Hence, we set it as a future job.

We hope that this work can inspire more research to investigate the under-studied image degradation.

## Acknowledgement

This work is jointly supported by the NSFC fund (61831014), in part by the Shenzhen Science and Technology Project under Grant (ZDYBH201900000002, JCYJ20180508152042002, CJGJZD20200617102601004).