# OpenReview forum: "Learning to Generate Realistic Noisy Images via Pixel-level Noise-aware  Adversarial Training"
_NeurIPS.cc/2021/Conference — NeurIPS 2021 Poster_

### Official Review · Reviewer_7ouL · 2021-07-08

**Rating:** 6
**Confidence:** 5

**Summary:**

This paper proposes a new method to simulate the real-world noisy images, which seems further improve the performance of current deep denoiser. A slight and effective generator is elaborately designed, and may facilitates the research of noise generation.
Besides, the proposed method combines lots of existing deep learning technologies, and surpasses the current SoTA by a large margin.

**Limitations And Societal Impact:**

The paper has addressed the limitations and potential negative societal impact in the supplementary material.

**Main Review:**

In summary, this work presents one noisy image generation method, and validates its effectiveness by comprehensive experiments to some extent.

1) As fas as I know, this is the first work to employ the pre-trained deep denoiser $D_d$ as a regularizer to constrain the generated noisy images. It is interesting, but not a general regularizer. In other words, I think such strategy will pull the generated noisy image distribution to that of the specific training data used to train $D_d$, which may not be desired by us in some cases.

2) The experiments only involve the comparison with other methods on the denoising task. For the noise generation task, this work does not cite and compare with some important works, such as:

     "Unprocessing Images for Learned Raw Denoising", CVPR 2019.

     "Dual Adversarial Network: Toward Real-World Noise Removal and Noise Generation", ECCV, 2020

3) In Table 4, the generator is trained on SIDD training dataset. After finetuning jointly on the SIDD training and the generated dataset, the performance on SIDD testing dataset has a little drop. So, the trained generator does not exactly capture the noise distribution of the training dataset (SIDD), even though with complex loss function?


**Time Spent Reviewing:**

4

---

> ### Author Response · Authors · 2021-08-08
> **Response to Reviewer 7ouL**
>
> Thanks for your valuable comments. Due to our negligence, we did not read [1] [2] before. We're sorry about that. We'll add citations and comparisons between PNGAN and similar methods, including the qualitative and quantitative results assessed by the metrics of [1]. Now we answer your questions one by one.
>
> (i) The generated noisy image distribution may be pulled to that of the specific training data.
>
> I agree with you. The problem you mentioned widely exists in the data-driven deep learning methods. No matter what model is used, during the training procedure, its function may easily tend to be pulled to a mapping from the input data distribution to ground-truth data distribution. This is a common limitation in deep learning caused by overfitting. For instance, in [1], DANet is firstly trained on SIDD. Subsequently, using DANet to perform data augmentation achieves significant improvement on SIDD and limited performance on Nam. To alleviate this limitation, this work adopts two strategies:
>
> Firstly, we set two hyper-parameters in synthesizing phase to adapt the initial distribution for different noisy scenes. For setting1, we use $\sigma_n$ to control the intensity of Gaussian distribution so as to accommodate different noise intensities. For setting2, the Poisson-Gaussian distribution can be approximated with a heteroscedastic Gaussian: $\sigma^2(L) = L \cdot \sigma_s^2 + \sigma_c^2$, where $ L \cdot \sigma_s^2$ denotes the signal-dependent noise component modeling photon sensing and $\sigma_c^2$ denotes the stationary noise component remaining stationary disturbances. The Poisson-Gaussian distribution is controlled by the hyper-parameter $L$. $L$ denotes the irradiance image of raw pixels. It can be derived by calculating the statistics of the noisy datasets so as to fit different real noisy scenes. In addition, the Poisson-Gaussian (setting2) is much closer to the distribution of real RAW noise [3] [4]. With this initial distribution, PNGAN can further narrow the discrepancy between the generated and real noisy images.
>
> Secondly, to promote the generalization ability of the finetuned models, we use large-scale HD datasets including DIV2K, Flickr2K, BSD68, Koadak24, and Urban100 to generate the extended noisy datasets so as to consider more real noisy scenes. However, although we adopt these strategies, PNGAN still can not accurately and completely model the distribution of real noise. Fortunately, the results reported in Tab. 1 suggest that PNGAN generalizes well across different denoising datasets and models.
>
> (ii) Discussions and comparisons with other noise generation methods.
>
> (1) Comparisons with DANet [1].
>
> Please refer to the response to the question (i) and (ii) of reviewer HLgB for comparisons between PNGAN and DANet [1].
>
> (2) Discussions with Unprocessing [2].
>
> Firstly, [2] proposes a technique for “unprocessing” generic images into data that resembles the raw measurements captured by real camera sensors, by modeling and inverting each step of a camera’s image processing pipeline. However, [2] doesn't propose a new method for noise generation. The raw noise modeling method used in [2] is also the Poisson-Gaussian distribution of traditional methods [3] [4]. Different from [2], we are not interested in the ISP pipeline modeling. Our method is dedicated to learning the final real noise representation rather than the intermediate form. We focus on the end-to-end mapping from synthetic noisy images (Gaussian or Poisson-Gaussian) to real noisy images. Specifically, we use a GAN and a pre-trained denoiser as the regularizer to establish a pixel-level adversarial training scheme that pulls the distribution of initial noise (Gaussian and Poisson-Gaussian) to that of real noise.
>
> Secondly, [2] is proposed for noise generation in RAW domain, where the noise is relatively simple and spatio-chromatically uncorrelated. PNGAN is proposed for generating realistic RGB noisy images, where the noise further affected by the ISP pipeline becomes more sophisticated and spatio-chromatically correlated.
>
> Thirdly, If using the technique propose in [2] to generate RGB noisy images, the Poisson-Gaussian noisy RAW image will pass through the hand-crafted ISP pipeline. This procedure may cause deviations of the generated RGB noisy images for two main reasons. Firstly, the Poisson-Gaussian distribution mainly considers two main noises, i.e., shot noise and read noise. However, there are other sources resulting in RAW noise. Thus, the Poisson-Gaussian distribution can not completely describe all the characteristics of RAW noise. Secondly, the ISP pipeline is very sophisticated and hard to be completely modeled. Each step of the ISP pipeline in [2] is hand-crafted and based on a specific mathematical assumption. This may lead to accumulated errors during the RAW2RGB and RGB2RAW transformation. Additionally, [2] mainly considers the mapping from clean RAW to clean RGB and its reversion. As for more sophisticated transformation, i.e., Poisson-Gaussian noisy RAW to real noisy RGB, [2] doesn’t analyze this situation and perform experiments to validate the effectiveness of the proposed technique. To alleviate these problems, in Setting2 of our work, we also select the Poisson-Gaussian noise as the initial distribution. After undergoing the pseudo ISP pipeline (RAW2RGB network), there’s also a domain discrepancy between the synthetic and real noisy images. Therefore, to narrow this discrepancy, PNGAN proposes a pixel-level adversarial training scheme to automatically pull the distribution of input noise to that of real camera noise. As reported in Tab. 2, when we directly use the synthetic data (Setting2) to train denoising models from scratch, there’s still a gap compared to those trained with real noisy data. After PNGAN is applied, the denoising models yield similar results with those trained with real noisy image pairs.
>
> Finally, using the technique of [2] to generate RGB noisy images requires the parameters of sensors and ISP components. However, these parameters are often unavailable. In this case, parameter adjusting will be a tedious and time-consuming problem. Besides, the components and sequence of the ISP pipeline of different devices may vary a lot. Using the same “unprocessing” strategy according to a specific camera and ISP pipeline to generate realistic noisy images across different datasets may cause undesired errors and hurt the quality of the generated noisy images. Due to these constraints, the extensibility and generalization ability of [2] may be limited. [2] validates its effectiveness by showing quantitative results of only a single denoising model on only a single denoising benchmark. In contrast, when using PNGAN to generate realistic noisy images, we don’t need to know the built-in parameters of the camera. This indicates that PNGAN has flexible extensibility. In addition, Tab. 1 of the main paper clearly suggests that PNGAN generalizes well across different denoising models and datasets.
>
> (iii) Explanation of "the little drop" and capturing SIDD noise distribution.
>
> The little drop of performance is not from the original model to the finetuned model. $q$ = 0 denotes finetuing the denoisers with only SIDD train set. In Line 340-347 of the paper, we evaluate the effect of the ratio of finetuning data. Thus, we denote the ratio of the extended training data to SIDD train set as $q$, i.e., $q$ = extended : SIDD. We change the value of $q$, finetune the original RIDNet, and test it on SIDD, PolyU, and Nam. We aim to achieve an optimal trade-off of performance between different datasets. Before being finetuned ($q$ = None in Tab. 4 denotes the original model), RIDNet only achieves 38.71 dB on SIDD. When $q$ = 0, all finetuning data comes from SIDD train set, in this case, the performance on SIDD achieves its maximum (39.32). However, the performance on the total dataset (38.92) is not good enough. Then we increase the value of $q$ to study its effect. The average performance on the three datasets yields the maximum (39.35) when $q$ = 60%. Thus, the finetuning phase doesn’t lead to a performance drop on SIDD comapred to the original model ($q$ = None, 38.71 dB). In contrast, no matter what the value of $q$ is, the performance of RIDNet obtains a reliable improvement.
>
> By the way, the little performance drop on SIDD from $q$ = 0 to $q$ = other values can be easily explained. When $q$ = 0, all the finetuning data comes from the SIDD dataset and strictly follows the distribution of SIDD. While $q$ = other values, part of the finetuning data comes from the five HD datasets (DIV2K, Flickr2K, BSD68, Koadak24, and Urban100). The discrepancy between different noisy datasets mainly consists of two parts: image and noise domain discrepancy. There's an image domain discrepancy between the extended HD dataset and SIDD clean dataset. There also exists an image domain discrepancy between the generated noisy data and SIDD noisy images. This domain discrepancy leads to the performance drop from $q$ = 0 to $q$ = other values on SIDD. Nonetheless, the total performance and the generalization ability of the finetuned model are improved. This doesn't indicate that PNGAN can't capture the distribution of SIDD noise. As reported in Tab. 2, when using setting1, PNGAN achieves 0.84 PGap on SIDD. When using setting2 to generate noisy images and then train the denoising models from scratch, the denoisers yield similar results compared to those trained with real noisy image pairs. This evidence shows that PNGAN is capable of capturing the distribution of SIDD.
>
> References:
>
> [1] "Dual Adversarial Network: Toward Real-World Noise Removal and Noise Generation", ECCV 2020
>
> [2] "Unprocessing Images for Learned Raw Denoising", CVPR 2019
>
> [3] "Practical poissonian-gaussian noise modeling and fitting for single-image raw-data",  TIP 2008
>
> [4] "Optimal inversion of the generalized anscombe transformation for poisson-gaussian noise", TIP 2012

---

> > ### Comment · Reviewer_7ouL · 2021-08-17
> > **Feedback to the author's responses**
> >
> > The authors have addressed most of my concerns. I agree to accept this paper.

---

### Official Review · Reviewer_NrP3 · 2021-07-09

**Rating:** 8
**Confidence:** 5

**Summary:**

The authors propose a pixel-level noise-aware generative adversarial network (PNGAN). They improve the denoising performance by generating more realistic noisy images.  Quantitative experiments prove that the performance of the existing denoising model can be significantly improved after being trained with the synthetic realistic noisy images.

**Ethical Concerns:**

No.

**Limitations And Societal Impact:**

There are some limitations that almost all deep learning methods will have, including the input dependence and performance limitations. The detailed limitations of the PNGAN are explained in the supplementary. This work has no limitations and potential negative social impact.

**Main Review:**

+ This work has many advantages, such as:

The authors solve certain limitations in the current real denoising task. For example, the domain difference between synthetic noise and real noise is alleviated. As a better noise data enhancement method, PNGAN reduces the difficulty of obtaining real noise image pairs.

The proposed method uses GAN to generate pixel-level synthetic noise, which is consistent with real-world noise. It can be used for effective training data augmentation.

Visually, PNGAN produces results that are closer to real noise. The quantitative experiments in Table 1 also show that the generated fake noisy images can be directly used for the training of the existing network, and the performance is improved by a large margin.

The paper is well-organized and clearly written, and the detailed source code is provided.

-	However, I still have the following concerns:

There is no quantitative evaluation of model parameters and interface speed in the paper.

For disordered and random noise, how to achieve Noise Domain Alignment in line140~175?

What are the differences and advantages of Pixel-level Noise Modelling and Image-level Noise Modeling?

Figure 4 shows that different data sets perform differently in domain discrepancy. Please analyze the reasons.

Finally, the method PNGAN is novel and its effectiveness is demonstrated by extensive ablation study and main comparisons. The authors can further improve the paper by trying to address the concerns.

-------------------------------------------------
All my concerns are well addressed in the rebuttal, and I would like to keep my initial rating.

**Time Spent Reviewing:**

16

---

> ### Author Response · Authors · 2021-08-07
> **Response to Reviewer NrP3**
>
> Thanks for your valuable comments. Now we respond to your concerns one by one.
>
> (i)  Quantitative evaluation of model parameters, computational cost, and inference speed.
>
> For noise generation, in Line 89 of the main paper, the Params of SIMNet (the generator) is 0.8M. When the input size is 256×256, the FLOPS of SIMNet is 83.2G. While testing SIMNet on a single RTX 8000 GPU at size 256×256, the inference speed is 26.67 fps. For noise removal, we directly adapt SOTA denoising models (MIRNet, MPRNet, and RIDNet). Therefore, the Params and FLOPS (at size 256×256) of the finetuning denoisers remain unchanged, as shown in the following table:
>
> | Methods | MIRNet | MPRNet | RIDNet |
> | :-----: | :----: | :----: | :----: |
> | Params (M) | 31.79 | 15.74 | 1.50 |
> | FLOPS (G) | 783.8 | 572.9 | 97.9 |
>
> (ii) For disordered and random noise, how to achieve noise domain alignment?
>
> As analyzed in Line 34-36 of the main paper, real camera noise is generally more sophisticated and signal-dependent. The noise produced by photon sensing is further affected by the in-camera signal processing (ISP) pipeline (e.g., Gama correction, compression, and demosaicing). Thus, due to the intrinsic randomness, complexity, and irregularity of real noise, the noise domain alignment can not be naively treated as a simple regression task. To model the distribution of real noise, we formulate the noise domain alignment into a maximum likelihood estimation problem in Eq. (3). To approach this maximum as close as possible, we propose a pixel-level adversarial training scheme between $G$ and $D$ that encourages $G$ to favor perceptually natural solutions residing on the manifold of real noisy images so as to construct the noise domain alignment.
>
> (iii) What are the differences and advantages of pixel-level noise modeling and image-level noise modeling?
>
> We analyze the differences between pixel- and image-level noise modeling as follow:
>
> (a) Image-level noise modeling treats a whole image as a random variable and outputs a value indicating how realistic this image is. Pixel-level noise modeling treats each pixel of a real noisy image as a random variable and outputs a score map. Each position of the score map indicates how realistic the corresponding noisy pixel is. As analyzed in Line 112-113 of the main paper, when taking multiple noisy images of the same scene, the noise intensity of the same pixel varies a lot. This indicates that each pixel on noisy images should be treated as a random variable. Consequently, image-level noise modeling may easily lead to coarse learning of real noise distribution while pixel-level noise modeling is more fine-grained and suitable for real noise scenes.
>
> (b) Additionally, as mentioned in Line 35-36, 113-114 of the main paper, the noise produced by photon sensing is further affected by the ISP pipeline and becomes spatio-chromatically correlated. However, image-level noise modeling treats images as samples and neglects the inter-pixel dependences, leading to a limited noise fitting. In contrast, pixel-level noise modeling implicitly learns the correlations between noisy pixels which benefits the realistic noise generation.
>
> (c) The ablation study in Line 324-326 of the main paper, Sec. 3, Fig. 3, and Tab. 1 of the supplementary shows that pixel-level noise modeling is more superior to image-level noise modeling in realistic noisy image generation.
>
> (iv) Why does PNGAN perform differently across different datasets in Fig. 4?
>
> To begin with, due to the difference in shooting scenes, imaging devices, illumination, camera movement, ISP pipelines, etc., there is an inevitable domain discrepancy between different real noisy datasets. Specifically, as analyzed in Line 239-243 of the main paper, the images in SIDD are collected using five smartphone cameras in 10 static scenes. The DND composes 50 noisy-clean image pairs captured by 4 consumer cameras. The PolyU dataset contains 40 different scenes captured by 5 cameras, including Canon EOS (5D Mark II, 80D, 600D), Nikon (D800), and Sony (A7 II). The Nam dataset is composed of real noisy images of 11 static scenes.  Thus, when using the same PNGAN to generate realistic noisy images, the decrement of domain discrepancy between generated and real noisy images may vary across different benchmarks. We measure the domain discrepancy between synthetic and real noisy datasets by treating fake and real noisy data as two distributions.  The metric Maximum Mean Discrepancy (MMD) and its detailed calculation process are described in Sec. 5 of the supplementary. The results in Figure 4 demonstrate that even for different data sets, our proposed method can effectively narrow the discrepancy between synthetic and real noisy datasets. It also shows that our method has good generalization for various real-world noises.

---

> > ### Comment · Reviewer_NrP3 · 2021-08-17
> > **Feedback to the author's responses**
> >
> > All my concerns are well addressed in the rebuttal, and I would like to keep my initial rating.

---

### Official Review · Reviewer_HLgB · 2021-07-13

**Rating:** 6
**Confidence:** 4

**Summary:**

This paper proposed a method to generate fake noisy-clean image pairs, which are expected to well simulate the real training pairs, in an adversarial training fashion, in order to augment the training set for a deep denoiser.

**Limitations And Societal Impact:**

The authors did not mention the limitations of their method. However, in my opinion, the major limitation is that it can not guarantee that the generated noisy images follow the same distribution of the real noisy images, although I know it is a very difficult problem.

**Main Review:**

The idea of adopt the adversarial training strategy to generate more training data is reasonable, and has been shown to be effective in previous studies. This paper has done good job in making this strategy also work in denoising problem, and shown promising results in experiments. The paper is also well-written and easy to follow.

The major problem of this work is that it misses discussion and comparison with a related study that addressed the similar problem. Specifically, Yue et al. proposed in [1] a dual adversarial training framework, which also aims to generate noisy-clean image pairs to augment the training set for real image denoising. The authors should discuss and empirically compare with this method. Besides, Yue et al. have also proposed two metrics to assess the performance of a real noisy image generator, and it is better to also consider these metrics in this work.

A minor problem: $\hat{I}_{clean}$  in the first equation of (1) should be  without "^" sign.

[1] Zongsheng Yue, Qian Zhao, Lei Zhang and Deyu Meng. Dual adversarial network: Toward real noise removal and noise generation. ECCV, 2020.

**Time Spent Reviewing:**

8 hours.

---

> ### Author Response · Authors · 2021-08-08
> **Response to Reviewer HLgB**
>
> Thanks for your valuable comments. Due to our negligence, we did not read [1] during the research. We're sorry about that. We will add citation and comparison with [1] in the revision, including the qualitative and quantitative results obtained by using the two assessment metrics of [1]. Now we answer your questions one by one.
>
> (i) Using the two metrics proposed in [1] to assess the performance of PNGAN
>
> We use the two metrics, PGap (PSNR Gap) and AKLD (Average KL Divergence) proposed in [1] to assess the performance and compare our method with previous algorithms. The results on SIDD are listed in the following table:
>
> | Methods | CBDNet | ULRD | GRDN | DANet | PNGAN (ours) |
> | :-----: | :----: | :----: | :----: | :----: | :----: |
> | PGap | 8.30 | 4.90 | 2.28 | 2.06 | 0.84 |
> | AKLD | 0.728 | 0.545 | 0.443 | 0.212 | 0.153 |
>
> Please note that the first four results are from Tab. 1 of [1]. In order to reduce the experimental errors, the PGap of PNGAN is derived by calculating the average PGap results of the three denoising models in Tab. 2 of the main paper, i.e., 0.84 = ((38.69 - 37.92)+(39.45 - 38.52)+(39.58-38.76))/3 = (0.77+0.93+0.82)/3. For fair comparisons, we use Gaussian noise the same with [1] as the initial input distribution. The calculating process of PGap and AKLD is also the same with [1]. It can be obviously observed that our PNGAN outperforms previous methods. These results suggest the superiority of PNGAN in realistic noisy image generation. In addition, We have performed visual examinations of our method and [1]. We observe that when using DANet to generate noisy images, it occasionally produces slight chessboard effects and black spots. However, using PNGAN has not encountered such a problem. Restricted by the rebuttal format, we can not provide the visual comparisons here. We are glad to add these quantitative and qualitative results to the revision.
>
>
> (ii) Discussion between DANet[1] and PNGAN
>
> To begin with, DANet [1] and PNGAN are both proposed for realistic noise generation and data augmentation for real denoising methods. However, they are different in:
>
> (a) DANet aims to learn the joint distribution of clean-noisy image pairs while PNGAN is dedicated to learning the real noise distribution. The learning target of PNGAN is more focused and requires less learning capacity.
>
> (b) DANet adopts image-level noise modeling while PNGAN uses pixel-level noise modeling. Image-level noise modeling treats a whole image as a random variable and outputs a value indicating how realistic this image is. Pixel-level noise modeling treats each pixel of a real noisy image as a random variable and outputs a score map. Each position of the score map indicates how realistic the corresponding noisy pixel is. As analyzed in Line 112-113 of the main paper, when taking multiple noisy images of the same scene, the noise intensity of the same pixel varies a lot. This observation indicates that each real noisy pixel should be treated as a sample. Therefore, image-level noise modeling may easily lead to coarse learning of real noise distribution while pixel-level noise modeling is more fine-grained and suitable for real noisy scenes. Additionally, as mentioned in Line 35-36, 113-114 of the main paper, the noise produced by photon sensing is further affected by the ISP pipeline and becomes spatio-chromatically correlated. However, image-level noise modeling treats images as samples and neglects the inter-pixel dependences, leading to a limited noise fitting. In contrast, pixel-level noise modeling implicitly learns the correlations between noisy pixels and thus benefits the realistic noise generation. As mentioned in Line 324-326 of the main paper, Sec.3 and Tab. 1 of the supplementary, pixel-level is much more effective than image-level in modeling the real noise distribution.
>
> (c) DANet starts from the Bayesian model and considers the conditional probability of both noise removal and generation perspectives. Specifically, as mentioned in Sec. 3.1 of [1], DANet introduces an additional latent variable z representing the fundamental elements conducting the hardware-related random noises. DANet assumes this latent variable can be easily set as an isotropic Gaussian distribution.  However, previous research [2] [3] reveals that the random noise distribution produced by photon sensors is much closer to the Poisson-Gaussian distribution. That is because sensor noise primarily comes from two sources: photon arrival statistics (“shot” noise) and imprecision in the readout circuitry (“read” noise). The Poisson part models the signal-dependent photon sensing while the Gaussian part remains the signal-independent stationary disturbances. Nonetheless, DANet overlooks this initial distribution (Poisson-Gaussian) and assumes that the latent hardware-related variable follows the Gaussian distribution. In contrast, PNGAN is motivated by the shooting situation of real noisy scenes and doesn’t base the noise model on any mathematical assumptions. We perform ablations by changing the initial distributions (Gaussian distribution, and Poisson-Gaussian distribution) to find out which input setting is more suitable for realistic RGB noisy image generation. As shown in Fig. 4 and Tab. 2 of the main paper, PNGAN with setting2 (Poisson-Gaussian initial distribution) generates more realistic noise patterns.
>
> (d) DANet proposes a dual adversarial training scheme, one is for pushing the distribution of the denoised images to that of clean images, the other is for pushing the distribution of the generated fake noisy images that of real noisy images. PNGAN is a single adversarial training scheme that relatively eases and stabilizes the training procedure and reduces the risk of model collapse. As mentioned in Line 126-127 of the main paper, PNGAN splits the realistic noisy image generation task into two sub-problems: image domain alignment and noise domain alignment.
>
> (e) Although DANet is an inspiring and interesting work, its generalization ability has not been further validated. Firstly, DANet mainly constructs experiments on the denoising models proposed in [1]. But for other SOTA methods such as RIDNet, [1] lacks experiments that using the generated noisy images to perform data augmentation. Secondly, although using DANet achieves a significant improvement on SIDD and DND datasets, the effectiveness of DANet on the Nam benchmark is limited. In contrast, as shown in Tab. 1 of the main paper, PNGAN generalizes well across different denoising models (MIRNet, MPRNet, and RIDNet) and datasets (SIDD, DND, Nam, and PolyU).
>
> (iii) Question about the notation in Eq. (1). What should the constant in Eq. (3) be set as? $\hat{I}_\{clean}$ or $I_\{clean}$?
>
> In fact, we set the constant in Eq. (1) as $\hat{I}_\{clean}$ but not ${I}_\{clean}$. $\hat{I}_\{clean}$ is the predicted counterpart of the real noisy image, $I_\{rn}$. It is derived by using $D_\{d}$ to denoise $I_\{rn}$ and not exactly the ground truth, i.e., $I_\{clean}$. We analyze why we split $I_\{rn}$ like this in Line 121-123. The mapping learned by $D_\{d}$ is precisely from $I_\{rn}$ to $\hat{I}_\{clean}$. If the constant component in Eq. (1) is set as $I_\{clean}$, the subsequent domain alignment will introduce unnecessary deviations and eventually lead to inaccurate results.
>
> (iv) Limitations of our work
>
> In fact, we analyze the limitations of our work in Sec. 10 of the supplementary. We agree with your comment that PNGAN can not guarantee that the generated noisy images follow the same distribution of the real noisy images. This is a common problem widely existing in data-driven deep learning methods. To alleviate this issue, we adopt two strategies. Please refer to the response to the question (i) of reviewer 7ouL for a detailed description. As reported in Tab. 1 of the main paper, a series of denoising models achieve a significant improvement on different datasets. This evidence demonstrates that our PNGAN generalizes well across different real denoising benchmarks and methods.
>
> References:
>
> [1] "Dual Adversarial Network: Toward Real-World Noise Removal and Noise Generation", ECCV 2020
>
> [2] "Practical poissonian-gaussian noise modeling and fitting for single-image raw-data", TIP 2008
>
> [3] "Optimal inversion of the generalized anscombe transformation for poisson-gaussian noise", TIP 2012

---

> > ### Comment · Reviewer_HLgB · 2021-08-17
> > **My concerns have been addressed**
> >
> > Thanks for your response. I think my concerns have been addressed.

---

### Official Review · Reviewer_oDMH · 2021-07-14

**Rating:** 7
**Confidence:** 4

**Summary:**

In this paper, the authors proposed a novel way to generate real-world noise, and presented the reliability of generating noise data sets on a large number of existing methods. Overall, the lack of real noisy images limits the performance of the fully supervised denoising network, and the proposed method contributes a reasonable solution.

**Limitations And Societal Impact:**

I think this method may have certain limitations in generalization, such as how to deal with unlearned real noise patterns in real-world applications.

**Main Review:**

The authors propose a simple but novel idea (PNGAN) for real image denoising. In my opinion, the motivation is reasonable by treating each noisy pixel as a random variable and then splitting noisy image generation into image and noise domain alignment. Compared with the previously proposed image-level GAN, the novel pixel-level PNGAN can obtain better predictions through the score map. The pixel-level adversarial training encourages the generator to pursue better solutions. In the experiments, ablation studies demonstrate the effects of PNGAN components, such as the proposed discriminator, pixel-level vs. image-level, components in the generator. Meanwhile, compared to existing works, the proposed PNGAN also achieved promising results.

However, I still have the following concerns:

(i) The novelty of the generator in Figure 3 is limited. Multi-scale attention and stacking of residual blocks are common components. But the use of fast 1D convolution is innovative.

(ii)It seems that Gaussian Poisson noise is better than Gaussian noise. The argument in favor of the used initial noise distribution should be provided.

(iii) What I am concerned about is whether the proposed PNGAN has good generalization?
(iv) The font in Figure 4 is too small, you can use abbreviations instead.
(v) In Fig 1 (a2), when the metadata of the image is unavailable, how does the PNGAN create noisy images using Setting2?
(vi) In Line 321-324 of the main paper, the authors claim that the image domain alignment is successfully conducted. It would be more convincing if more results are provided.





**Time Spent Reviewing:**

8

---

> ### Author Response · Authors · 2021-08-04
> **Response to Reviewer oDMH**
>
> Thanks for your valuable comments, now we respond to your concerns one by one.
>
> (i) Contributions of our work and motivations of using the multi-scale attention mechanism.
>
> The main contributions of this work are the simple yet reasonable noise model and PNGAN framework for realistic noisy image generation. Specifically, the noise model treats each noisy pixel as a random variable and then splits the noisy image generation into two parts: image and noise domain alignment. The PNGAN framework establishes an effective pixel-level adversarial training scheme to encourage the generator to favor solutions that reside on the manifold of real noisy images. The motivation for using the multi-scale mechanism is detailed in Line 203-208 of the main paper. We adopt 1D convolutional attention block to replace 2D channel attention module for more efficient model inference. The ablation study in Tab. 3 and Line 331-334 shows that the multi-scale attention mechanism effectively improves the quality of the generated noisy images.
>
> (ii)​ Why do we select Gaussian and Poisson-Gaussian as the initial noise distribution? Why is Poisson-Gaussian noise better?
>
>  We adopt Gaussian and Poisson-Gaussian distribution as the initial setting for the following reasons. To begin with, many previous denoising methods are performed with adding AWGN to synthesize noisy-clean image pairs. Besides, a large number of researches in image synthesis show that the normal distribution has the potential to simulate any data distribution. Thus, setting1 naively and empirically adopts Gaussian distribution as input.
>
> However, as analyzed in Line 33-37, the real-camera noise is signal-dependent and much more sophisticated than AWGN. To bridge the huge gap between real noise and AWGN, we need to customize another initial distribution that is closer to real noise.
>
> Therefore, setting2 adopts Poisson-Gaussian distribution as input, which models the signal-dependent photon sensing with Poisson and remains signal-independent stationary disturbances with Gaussian. This initial noise distribution is much closer to the shot noise and read noise in the real-camera RAW domain as analyzed in [1]. Additionally, the Gaussian distribution is a normal distribution, while the Poisson distribution is the discrete counterpart of the Gaussian distribution, which has high randomness. The real-world noise is also distributed irregularly and randomly. Therefore, the Poisson-Gaussian model can better fit the real noise [2] [3]. As a result, the generated fake noise with setting2 is much more similar to the real noise.
>
> (iii) The generalization ability of PNGAN.
>
> We note that different denoising models have different learning capacities. Simultaneously, there remains a domain discrepancy between different noisy datasets due to the imaging devices and shooting scenes. Specifically, as analyzed in Line 239-243 of the main paper, the images in SIDD are collected using five smartphone cameras in 10 static scenes. The DND composes 50 noisy-clean image pairs captured by 4 consumer cameras. The PolyU dataset contains 40 different scenes captured by 5 cameras, including Canon EOS (5D Mark II, 80D, 600D), Nikon (D800), and Sony (A7 II). The Nam dataset is composed of real noisy images of 11 static scenes. Thus, to demonstrate the generalization ability of our method, we use PNGAN to create noisy images on four real denoising benchmarks (SIDD, DND, PolyU, and Nam) and then perform data augmentation for different denoising models (MIRNet, MPRNet, and RIDNet). As shown in Fig. 5 of the main paper and the supplementary, the intensity and distribution of the generated noise are visually similar to those of real noise. Then we use the created realistic noisy images to perform data augmentation, a series of real denoising models are improved significantly as shown in Tab. 1, Fig. 6 of the main paper, and Fig. 7 of the supplementary. In particular, as listed in Tab. 1, RIDNet* is 0.54, 0.29, 0.68, and 0.49 dB higher than RIDNet on SIDD, DND, PolyU, and Nam. MPRNet* achieves 0.35, 0.38, 1.41, and 1.31 dB gain than MPRNet. MIRNet* is improved by 0.35, 0.35, 1.37, and 1.21 dB. These results clearly suggest that our PNGAN generalizes well across different real denoising benchmarks and algorithms.
>
> (iv) Font size in Fig. 4.
>
> We will use abbreviations and larger font in the revision to ensure that the text information in the figures can be clearly seen.
>
> (v) How do we use setting2 to produce noisy images when the RAW data is unavailable?
>
> In Line 42-43 of the main paper, we use a pre-trained RGB2RAW network the same with CycleISP to transform the clean RGB image to its RAW counterpart. Subsequently, we utilize this RAW image to create realistic noisy images with setting2.
>
> (vi) More experimental results to validate the image domain alignment.
>
> Firstly, in Line 126, the image domain alignment aims to align $\hat{I}_\{clean}$. Then in Line 321-324 of the main paper, Sec.6 and Tab. 2 of the supplementary, we validate the image domain alignment by evaluating the PSNR and SSIM between $D_d({I_\{fn}})$ and $D_d({I_\{rn}})$. The PSNR and SSIM between the denoised counterparts of the generated and real noisy images are 39.14 dB and 0.928 on average respectively. Detailed results of different real denoising models are listed in Tab. 2 of the supplementary. These results demonstrate that the image domain alignment is conducted.
>
> (vii) Limitations in generalization. How to deal with unlearned real noise patterns in real-world applications？
>
> Firstly, we analyze the limitations of our work in Sec. 10 of the supplementary. In fact, PNGAN can not guarantee that the generated noisy images follow the same distribution of the real noisy images. This is a common problem widely existing in data-driven deep learning methods. To alleviate this issue, we adopt two strategies. Please refer to the response to the question (i) of reviewer 7ouL for a detailed description. As shown in Tab. 1 of the main paper, a series of denoising models achieve a significant improvement on different datasets. This evidence clearly suggests that our PNGAN generalizes well across different real denoising benchmarks and methods. Please refer to question (iii) for a more detailed analysis.
>
> References:
>
> [1] A. J. Blanksby, M. J. Loinaz, D. A. Inglis and B. D. Ackland, "Noise performance of a color CMOS photogate image sensor", Proc. IEEE Int. Electron Devices Meet. Dig., pp. 205-208, 1997
>
> [2] Alessandro Foi, Mejdi Trimeche, Vladimir Katkovnik, and Karen Egiazarian. Practical poissonian-gaussian noise modeling and fitting for single-image raw-data. TIP, 2008. 2
>
> [3] Markku Makitalo and Alessandro Foi. Optimal inversion of the generalized anscombe transformation for poisson-gaussian noise. TIP, 2012. 6

---

### Decision · Program_Chairs · 2021-09-27

**Decision:**

Accept (Poster)

**Comment:**

This paper proposes a novel idea to synthesize real world noisy images for image denoising.
The main idea is to treat each noisy pixel as a random variable and then split noisy image generation into image domain alignment and noise domain alignment. Based on the idea, a GAN method is proposed to perform the pixel level adversarial training. The idea is interesting and novel, and the effectiveness of the proposed method is demonstrated in the experiments by comparing it with SOTA methods and detailed ablation studies. All the reviewers are also impressed by the idea and the promising experimental results. However, a major drawback of the work is the lack of theoretical guarantee on the distribution gap between the real noisy images and the generated noisy images. Given that this is the first work that moves towards real world noisy image generation,  I recommend acceptance given the novelty and the practical significance of the proposed work.